# Rethinking the Temporal Modeling For Time Series Forecasting with Hybrid Modeling

## Abstract

Time series forecasting is a critical task in various domains, including traffic, energy, and weather series forecasting. Recent research has explored the utilization of MLPs, Transformers, and CNNs architectures for time series modeling, delivering promising results. In this work, we take a step further by systematically studying the strengths and limitations of these methods and integrating their capabilities to formulate a unified framework for time series forecasting with a hybrid modeling approach. We introduce UniTS, a simple yet scalable framework for temporal modeling that incorporates multiple feature learning techniques. Moreover, prior research employed different hyperparameter configurations in various temporal modeling approaches, which might causing unfair performance comparisons. For instance, when predicting with the same forecasting horizon, prior approaches might exhibit significant variations in lookback window lengths. In our study, we address this issue by validating and standardizing parameters that can significantly impact performance, ensuring a more equitable comparison of models across diverse datasets. UniTS achieves state-of-the-art performance across various domains, and we conduct extensive experiments to further evaluate its capabilities. Our results are fully reproducible, and the source code for this work is available at `https://anonymous.4open.science/r/UniTS-8DA8/README.md`.

## 1 Introduction

Time series forecasting task plays a pivotal role in our daily lives, influencing decisions ranging from weather forecasting, energy consumption management, urban traffic forecasting to stock market predictions (Barrera-Animas et al., 2022; Wang et al., 2023; Sezer et al., 2020; Shaikh et al., 2022). Accurate predictions enable us to anticipate future trends, allocate resources efficiently, and make informed choices. In the realm of machine learning, designing effective frameworks for time series analysis is of paramount importance. These frameworks, being categorized into distinct types, are instrumental in extracting meaningful insights from temporal data. In this paper, we group them into three primary categories: (1) Models use Linear layers only, represented by the influential model DLinear (Zeng et al., 2023); (2) Transformer-like models, exemplified by the model PatchTST (Nie et al., 2022); (3) Models use Convolutional Neural Networks (CNNs), like MICN and TimesNet (Wang et al., 2022; Wu et al., 2022).

It's worth noting that these categories are not rigidly isolated but rather represent key approaches within the broader landscape of time series analysis. To illustrate this point, let's delve into two examples: the Multilayer Perceptrons (MLPs) and CNNs. MLPs operates in a holistic fashion by processing the entire series sequence through linear layers, granting access to global information. Recently, DLinear (Zeng et al., 2023) has demonstrated that a single-layer linear network is sufficient to train an accurate time series forecasting linear network, and the model's performance is significantly better than previous models that employed attention mechanisms and encoder-decoder architectures (Zhou et al., 2021; 2022; Zeng et al., 2023; Wu et al., 2021). On the other hand, models like MICN and TimesNet (Wang et al., 2022; Wu et al., 2022), employing CNNs, primarily focus on extracting features from smaller partitions of the sequence using convolutional kernels with specific lengths and strides. Beyond to these two types of models, Transformer models (Wu et al., 2021; Zhou et al., 2021; 2022) have also been proposed for application in time series forecasting, with PatchTST being a representative example (Nie et al., 2022). PatchTST's prediction module includes

a linear layer for the prediction projection similar to the DLinear. However, compared to DLinear, which directly maps from input sequences to prediction sequences, PatchTST incorporates modules such as attention layers, layer normalization, linear layers, and positional embeddings before the linear mapping.

In summary, MLPs and Transformers handle global information effectively with a larger receptive field, while CNNs focus on local features due to limited receptive field, indicating that they are not entirely mutually exclusive. In general, single modeling design can lead to performance limitations. For example, in previous experiments (Wu et al., 2022; Wang et al., 2022; Nie et al., 2022; Zeng et al., 2023), shorter lookback windows favor CNNs over MLPs and Transformers, but as the window size increases, CNNs struggle to capture complete sequences while the other models perform well. This leads us to consider hybrid modeling approaches to combine the strengths of both. However, another significant challenge lying in designing and validating time series forecasting models is the difficulty of aligning experimental results of existing work, since prior studies employed different hyperparameter configurations (Zhou et al., 2021; Wu et al., 2021; Wang et al., 2022; Zeng et al., 2023; Nie et al., 2022; Wu et al., 2022). For instance, when predicting with the same forecasting horizon, these approaches might exhibit significant variations in lookback window length, which could be a crucial factor resulting in unfair performance comparisons. Nevertheless, our experiments highlight the significance of comprehending the interplay between the lookback window length and model performance.

Motivated by the aforementioned issues, in this work, we enhance our understanding of contemporary time series forecasting methods and their practical applicability. We compare representative methods using MLPs, CNNs, and Transformers, addressing alignment issues by standardizing critical parameters for fair model comparisons across diverse datasets. We further introduce "UniTS", a novel time series hybrid modeling framework that integrates these methods. In summary, our contributions are as follows:

1. We delve into an in-depth exploration of the significance of modules used in prior studies, unveiling numerous insightful phenomena that inspire the design of temporal modeling for time series forecasting.

2. We propose a simple, scalable, and integrated machine learning architecture that combines multiple learning modes, providing a unified framework for time series analysis. The proposed model achieves state-of-the-art performance across various datasets.

3. We address the issue of misalignment in previous work settings by validating and standardizing parameters that can significantly impact performance, thereby ensuring a more equitable comparison of models across diverse datasets. Additionally, we carried out network architecture parameter search experiments under a limited search budget, highlighting the research potential in network and parameter exploration for time series forecasting tasks.

## 2 RELATED WORK

The design of machine learning frameworks for time series analysis has garnered significant attention in recent years. In this section, we provide an overview of related work, categorizing existing approaches and highlighting their key characteristics.

**MLPs for Time Series Forecasting**   Employing linear layers is a fundamental approach to time series forecasting (Zeng et al., 2023; Das et al., 2023; Xu et al., 2023; Li et al., 2023). The most representative among these studies is the work of DLinear (Zeng et al., 2023), which purely relies on linear transformations to extract temporal features. This approach allows for direct processing of sequences through linear layers, granting access to global information.

**CNNs for Time Series Forecasting**   CNNs have emerged as a powerful tool for time series analysis Sen et al. (2019); Hewage et al. (2020). Recently, models like MICN (Wang et al., 2022) and TimesNet (Wu et al., 2022) exemplify the CNN-based approaches, which emphasize the extraction of local features from smaller sequence partitions using convolutional kernels with specific lengths and strides, showcasing excellent performance in forecasting tasks.

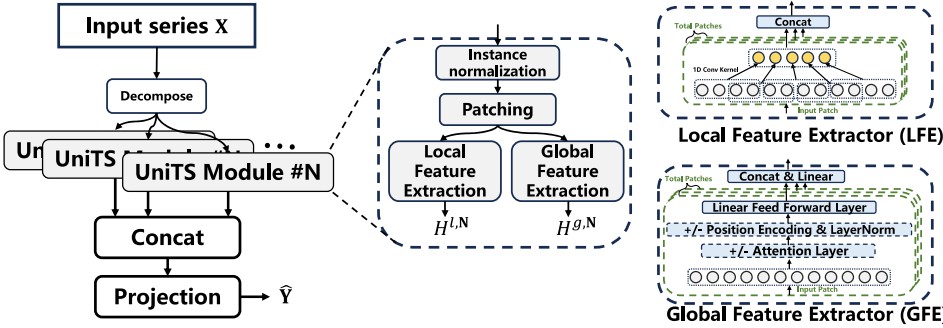

Figure 1: An overview of the proposed framework "UniTS", $H^{l,N}$ and $H^{g,N}$ stand for the global feature embeddings and local feature embeddings extracted from UniTS Module#$N$.

**Transformers for Time Series Forecasting**    In recent years, Transformer-like models have gained prominence in various domains (Vaswani et al., 2017; Radford et al., 2021; Han et al., 2022; Touvron et al., 2023). A series of methods (Zhou et al., 2021; Wu et al., 2021; Zhou et al., 2022; Nie et al., 2022) have extended the Transformer architecture to time series analysis. The Transformer layer comprises components including attention layers, layer normalization, linear layers, and positional embeddings. Importantly, these components can be independently integrated into the network, allowing for a flexible exploration of their impact on predictive performance.

## 3 HYBRID TEMPORAL MODELING FOR TIME SERIES FORECASTING

In this section, we first introduce the problem formulation throughout this paper. Then, we introduce temporal modeling modules and techniques, and illustrate how we have utilize these methods to construct a hybrid temporal modeling framework "UniTS".

### 3.1 PROBLEM STATEMENT

Given a set of series with the same length of lookback window length, our objective is to utilize a model to predict a multivariate time series of given prediction horizons. To clarify, we are tasked with forecasting future data points in a multivariate time series based on historical observations. Formally, we denote: $C$: The number of series, $L$: The length of lookback window length, $T$: The length of the prediction horizons, $\mathbf{X}_i = \{x^i_1, \cdots, x^i_L\}$: The input time series sequence for the $i$-th series, where $i \in [1, C]$, $\mathbf{Y}_i = \{x^i_{L+1}, \cdots, x^i_{L+T}\}$: The predicted time series for the $i$-th series. For each time series sequence $\mathbf{X}_i$ in the dataset, we input it into the neural network model, which consists of multiple layers of neurons, to obtain a prediction $\hat{\mathbf{Y}}_i$, yielding a mapping: $\hat{\mathbf{Y}}_i = f(\mathbf{X}_i)$. Here, $\hat{\mathbf{Y}}_i$ represents the ground truth prediction for the $i$-th series, and $f$ denotes the neural network model that captures the underlying patterns and dependencies in the input time series $\mathbf{X}_i$ to generate the forecasting values.

### 3.2 PROPOSED FRAMEWORK

The overall framework of UniTS can be seen from Figure 1. Firstly, it will preproess the data. Then, the preprocessed data is fed into both the global feature extractor and the local extractor. Finally, the output embeddings will be concatenated and fed to a final projection layer to get the final prediction.

### 3.2.1 DATA PREPROCESSING

Data preprocessing is of critical importance in context of the temporal modeling for time series forecasting (Wu et al., 2022; Zeng et al., 2023; Nie et al., 2022; Das et al., 2023; Wang et al., 2022). Primarily, the data preprocessing process in the recent studies can be summarized into 3 parts: 1) Series Decomposition; 2) Data patching, and 3) Instance Normalization.

**Series Decomposition**  Series decomposition was first employed in Autoformer (Wu et al., 2021), which first employed a seasonal-trend decomposition step before each neural block. It is a conventional technique in time series analysis aimed at enhancing the predictability of raw data. Specifically, the vanilla series decomposition module utilizes a moving average kernel on the input sequence to extract the trend-cyclical component of the time series. The difference between the original sequence and the extracted trend component is considered as the seasonal component. Such design has been widely applied in the recent stuides (Wu et al., 2021; Zhou et al., 2022; Zeng et al., 2023; Wang et al., 2022; Das et al., 2023), and showed its effectiveness in case studies. Built upon the decomposition approach of Autoformer, FEDformer (Zhou et al., 2022) introduces a strategy involving the mixture of experts to combine trend components extracted using moving average kernels with varying kernel sizes. In subsequent work, as demonstrated in MICN (Wang et al., 2022), the authors implemented a design that averages multiple kernel sizes within the decomposition module. UniTS also incorporates a series decomposition module as a component of the model.

**Patching**  Data patching is an potent strategy to mitigate computational complexity, particularly in scenarios with an extensive lookback window or when handling voluminous datasets. Given an input time series $\mathbf{X}_i$, the data patching approach involves an initial partitioning into patches, with the flexibility to configure them as either overlapping or non-overlapping. Let $P$ denote the patch length, and $S$ represent the stride, defining the non-overlapping interval between consecutive patches. The patching procedure generates a sequence of patches, $\mathbf{X}_i^P \in \mathbb{R}^{N \times P}$, where $N$, the number of patches, is computed as $N = \lfloor \frac{(L-P)}{S} \rfloor + 2$. In this context, $S$ repetitions of the final value, denoted as $x_L^i \in \mathbb{R}$, are appended to the end of the original sequence before patching. Leveraging patches enables a reduction in the number of input tokens from $L$ to approximately $L/S$. For temporal modeling methods employing transformer or linear structures, this reduction signifies a quadratic decrease in both memory usage and computational complexity of long sequence by a factor of $S$ (Nie et al., 2022).

Recent work PatchTST (Nie et al., 2022) and TimesNet (Wu et al., 2022) have used two different data patching methods. PatchTST employs a predefined patch size for segmentation, while TimesNet uses periodicity for selecting dominant frequency components in the frequency domain. By obtaining the primary frequency components, it performs data patching of input sequence with their corresponding periods. However, TimesNet's approach requires sequential computations and operate Fourier transform computations for each batch, causing significantly higher computational complexity. In consideration of the aforementioned factors, we adopt PatchTST's patch segmentation strategy for performance evaluation, which is simpler and more effective in experiments.

**Instance Normalization**  Instance Normalization is a technique introduced to address the distribution shift effect between training and testing data (Ulyanov et al., 2016; Kim et al., 2021), which plays a pivotal role in our approach. This method involves the straightforward normalization of each time series instance by subtracting the mean and dividing by the standard deviation. Essentially, we apply this normalization to each time series before the patching process, and subsequently, the mean and deviation are reintegrated into the output prediction. In this study, we also incorporate the ReVIN method (Kim et al., 2021) to project the normalized batch samples into a learnable parameterized distribution. After patching and data instance normalization, the processed data would be fed into global feature extractors and local feature extractors.

### 3.2.2  GLOBAL FEATURE EXTRACTION

For the global feature extractor, we follow the design used in DLinear (Zeng et al., 2023) by utilizing a simple variant of the direct multi-step forecasting model (Chevillon, 2007). It constructs a competitive baseline by employing only one linear layer for time series forecasting. In this work, we first pass the patch through a linear preceding layer, as illustrated in the right bottom of Figure 1, mapping each patch to $d$ dimensions. Furthermore, stacking multiple linear layers here further provides a remedy for the potential underfitting issue in real-world scenarios, which might occur as a consequence of a shallow structure. Moreover, the introduction of attention, position encoding, and layernorm mechanisms can further transform it into a comprehensive transformer layer. We will delve into a more detailed discussion of the impact of this structural design on performance in Section 4.4. Subsequently, embeddings of all patches are concatenated, followed by passing through a linear layer to obtain the final global latent embedding of dimension $d$.

### 3.2.3 LOCAL FEATURE EXTRACTION

The main network architecture of the local feature extractor is CNNs, with each layer consisting of serveral convolutional kernels of different scales tailored to model distinct temporal patterns by utilizing various scale sizes. The local feature extractor module plays a pivotal role in capturing the local features of the given sequence from a multi-scale view. To elaborate, the local feature extraction module performs 1D convolution on patched sequence for downsampling. This process can be represented as: $)h_{j,i}^{l,k} = \text{Conv1D}_j(\text{Padding}(h_{j,i}^{l,k-1}))$, where $h_{j,i}^l$ represents the local latent embedding of series $i$ within 1D convolution kernel $k$ at layer $l$. The final local feature is obtained by concatenating all the convolved features, represented as $H_i^l = \text{Concat}(h_{1,i}, \cdots, h_{K,i})$. Insipred by (Wang et al., 2022), we can utilize multiple kernels to capture temporal patterns across various scale sizes.

### 3.2.4 PREDICTION PROJECTION

Given the extracted global feature embeddings $H^g = \{H^{g,1}, \cdots, H^{g,N}\}$ and the local feature embeddings $\{H^{l,1}, \cdots, H^{l,N}\}$, we concatenate them and use a linear projection layer to get the final prediction: $\hat{\mathbf{Y}}_{\mathbf{i}} = \text{Projection}(\text{Concat}(H_i^g, H_i^l))$.

## 4 EXPERIMENTS

### 4.1 EXPERIMENTAL SETUP

**Datasets** The performance evaluation of our experiments encompasses eight widely recognized datasets including Weather, Traffic, Electricity, ILI, and four ETT datasets (ETTh1, ETTh2, ETTm1, ETTm2). These datasets have served as prevalent benchmarks and are publicly accessible via the work of (Wu et al., 2021). The summarization of dataset statistics can be found in Appendix A.1.

**Baselines** Our selection of state-of-the-art Transformer-based models for baselines includes PatchTST Nie et al. (2022), FEDformer (Zhou et al., 2022), Autoformer Wu et al. (2021), and linear model DLinear (Zeng et al., 2023). Additionally, we include two CNN-based models, MICN (Wang et al., 2022) and TimesNet (Wu et al., 2022), and one RNN-based model: LSTNet (Lai et al., 2018). Additionally, in Appendix 5, we have provided supplementary experimental results and performance analyses for recent proposed models that have been demonstrated to be effective for time series prediction. This includes TiDE (Das et al., 2023), N-BEATS (Oreshkin et al., 2019), N-HiTS (Challu et al., 2023), SpaceTimeFormer (Grigsby et al., 2021), and RLinear/RMLP (Li et al., 2023).

**Experiment Settings** All models adhere to a consistent experimental setup, with prediction length denoted as $T$, where $T$ takes on values of 24, 36, 48, 60 for the ILI dataset and 96, 192, 336, 720 for other datasets, as specified in the previous works (Wu et al., 2021; Zhou et al., 2022; Nie et al., 2022; Zeng et al., 2023). Notably, in work like (Wu et al., 2021; Zhou et al., 2021; 2022; Wu et al., 2022), they use a fixed lookback window for performance evaluation. However, other studies like (Wang et al., 2022; Zeng et al., 2023; Nie et al., 2022) finetunes the lookback horizon as a hyperparameter for the performance evaluation. This distinction makes it difficult to directly compare their performance, however, we believe that both of these settings are meaningful, and the performance differences between different models in these two settings further prompt us to consider how the lookback window length affects the model's predictive performance. Therefore, in the experiments conducted in this paper, we have designed two types of experiments based on the distinction between these two settings: (i) Prediction performance without a constraint on lookback window length; (ii) Prediction performance with a fixed lookback window length. For setting (i), baseline results are collected by running experiments with six different lookback window lengths $L \in \{36, 60, 84, 108\}$ for ILI and $L \in \{96, 288, 384, 576, 640, 720\}$ for the other five datasets. The best results among these configurations are chosen as the performance metric. For setting (ii), baseline results are obtained by running experiments with a fixed lookback window length. For the evaluation of multivariate time series forecasting, we employ Mean Squared Error (MSE) and Mean Absolute Error (MAE) as our metrics.

Table 1: Multivariate long-term forecasting results are evaluated using prediction length $T \in \{24, 36, 48, 60\}$ for the ILI dataset and $T \in \{96, 192, 336, 720\}$ for other datasets. A lower MSE or MAE indicates a better prediction. The best results are highlighted in **bold**, the second best is underlined.

| Models | | UniTS | | PatchTST | | DLinear | | TimesNet | | MICN | | Fedformer | | Autoformer | | LSTNet | |
|---|---|---|---|---|---|---|---|---|---|---|---|---|---|---|---|---|---|
| Metric | | MSE | MAE | MSE | MAE | MSE | MAE | MSE | MAE | MSE | MAE | MSE | MAE | MSE | MAE | MSE | MAE |
| ETTh1 | 96 | **0.365** | **0.390** | _0.370_ | _0.399_ | 0.384 | 0.399 | 0.384 | 0.400 | 0.391 | 0.420 | 0.376 | 0.419 | 0.435 | 0.446 | 1.044 | 0.773 |
| | 192 | **0.396** | **0.411** | _0.412_ | _0.418_ | 0.412 | 0.420 | 0.430 | 0.445 | 0.430 | 0.453 | 0.420 | 0.448 | 0.456 | 0.457 | 1.217 | 0.832 |
| | 336 | **0.414** | **0.426** | _0.422_ | _0.440_ | 0.443 | 0.448 | 0.479 | 0.460 | 0.439 | 0.458 | 0.459 | 0.465 | 0.486 | 0.487 | 1.259 | 0.841 |
| | 720 | **0.418** | **0.435** | _0.447_ | _0.453_ | 0.501 | 0.490 | 0.513 | 0.489 | 0.485 | 0.502 | 0.506 | 0.507 | 0.515 | 0.517 | 1.271 | 0.838 |
| ETTh2 | 96 | **0.262** | **0.329** | _0.269_ | _0.336_ | 0.289 | 0.353 | 0.335 | 0.369 | 0.331 | 0.375 | 0.358 | 0.397 | 0.332 | 0.368 | 2.522 | 1.278 |
| | 192 | **0.327** | **0.360** | _0.339_ | _0.379_ | 0.376 | 0.411 | 0.397 | 0.405 | 0.419 | 0.437 | 0.429 | 0.439 | 0.426 | 0.434 | 3.312 | 1.384 |
| | 336 | **0.349** | **0.384** | _0.361_ | _0.397_ | 0.435 | 0.442 | 0.449 | 0.450 | 0.442 | 0.468 | 0.496 | 0.487 | 0.477 | 0.479 | 3.291 | 1.388 |
| | 720 | **0.377** | **0.415** | _0.379_ | _0.422_ | 0.467 | 0.479 | 0.451 | 0.455 | 0.439 | 0.462 | 0.463 | 0.474 | 0.453 | 0.490 | 3.257 | 1.357 |
| ETTm1 | 96 | **0.289** | **0.343** | _0.291_ | _0.346_ | 0.303 | 0.348 | 0.330 | 0.365 | 0.354 | 0.393 | 0.379 | 0.419 | 0.510 | 0.492 | 0.863 | 0.664 |
| | 192 | **0.326** | **0.360** | _0.334_ | _0.364_ | 0.335 | _0.364_ | 0.370 | 0.382 | 0.400 | 0.412 | 0.426 | 0.441 | 0.514 | 0.495 | 1.113 | 0.776 |
| | 336 | **0.355** | **0.382** | _0.367_ | 0.387 | 0.368 | _0.386_ | 0.402 | 0.400 | 0.401 | 0.408 | 0.445 | 0.459 | 0.510 | 0.492 | 1.267 | 0.832 |
| | 720 | **0.406** | **0.408** | _0.415_ | _0.420_ | 0.420 | 0.421 | 0.468 | 0.442 | 0.445 | 0.450 | 0.543 | 0.490 | 0.527 | 0.493 | 1.324 | 0.858 |
| ETTm2 | 96 | **0.159** | **0.250** | _0.164_ | _0.256_ | 0.166 | 0.259 | 0.180 | 0.258 | 0.202 | 0.284 | 0.203 | 0.287 | 0.205 | 0.293 | 2.041 | 1.073 |
| | 192 | **0.213** | **0.285** | _0.220_ | _0.296_ | 0.223 | 0.302 | 0.244 | 0.302 | 0.261 | 0.324 | 0.269 | 0.328 | 0.278 | 0.336 | 2.249 | 1.112 |
| | 336 | **0.264** | **0.311** | _0.273_ | _0.329_ | 0.281 | 0.340 | 0.311 | 0.339 | 0.302 | 0.348 | 0.325 | 0.366 | 0.343 | 0.379 | 2.568 | 1.238 |
| | 720 | **0.344** | **0.353** | _0.359_ | _0.385_ | 0.389 | 0.413 | 0.405 | 0.401 | 0.385 | 0.400 | 0.421 | 0.415 | 0.414 | 0.419 | 2.720 | 1.287 |
| Traffic | 96 | **0.355** | **0.243** | _0.360_ | _0.249_ | 0.398 | 0.269 | 0.573 | 0.302 | 0.515 | 0.307 | 0.587 | 0.366 | 0.597 | 0.371 | 0.843 | 0.453 |
| | 192 | **0.365** | **0.251** | _0.381_ | _0.256_ | 0.412 | 0.284 | 0.601 | 0.333 | 0.535 | 0.314 | 0.604 | 0.373 | 0.607 | 0.382 | 0.847 | 0.453 |
| | 336 | **0.380** | **0.260** | _0.394_ | _0.264_ | 0.420 | 0.291 | 0.615 | 0.338 | 0.526 | 0.310 | 0.621 | 0.383 | 0.623 | 0.387 | 0.853 | 0.455 |
| | 720 | **0.417** | **0.279** | _0.427_ | _0.282_ | 0.454 | 0.307 | 0.628 | 0.345 | 0.569 | 0.321 | 0.626 | 0.382 | 0.639 | 0.395 | 1.500 | 0.805 |
| Electricity | 96 | **0.130** | **0.221** | **0.130** | _0.222_ | 0.140 | 0.237 | 0.167 | 0.271 | 0.191 | 0.303 | 0.193 | 0.308 | 0.196 | 0.313 | 0.375 | 0.437 |
| | 192 | **0.146** | **0.241** | _0.148_ | _0.248_ | 0.158 | 0.254 | 0.184 | 0.285 | 0.199 | 0.306 | 0.201 | 0.315 | 0.211 | 0.324 | 0.442 | 0.473 |
| | 336 | **0.159** | **0.256** | _0.163_ | _0.259_ | 0.174 | 0.276 | 0.197 | 0.298 | 0.215 | 0.321 | 0.214 | 0.329 | 0.214 | 0.327 | 0.439 | 0.473 |
| | 720 | **0.195** | **0.292** | _0.197_ | _0.294_ | 0.208 | 0.313 | 0.218 | 0.315 | 0.221 | 0.326 | 0.246 | 0.355 | 0.236 | 0.342 | 0.980 | 0.814 |
| Weather | 96 | **0.144** | **0.189** | _0.150_ | _0.203_ | 0.168 | 0.210 | 0.172 | 0.219 | 0.182 | 0.249 | 0.217 | 0.296 | 0.249 | 0.329 | 0.369 | 0.406 |
| | 192 | **0.187** | **0.240** | _0.193_ | _0.241_ | 0.208 | 0.254 | 0.217 | 0.261 | 0.243 | 0.312 | 0.276 | 0.336 | 0.325 | 0.370 | 0.416 | 0.435 |
| | 336 | **0.236** | **0.271** | _0.247_ | _0.282_ | 0.263 | 0.321 | 0.277 | 0.304 | 0.291 | 0.332 | 0.339 | 0.380 | 0.351 | 0.391 | 0.455 | 0.454 |
| | 720 | **0.303** | **0.320** | _0.316_ | _0.334_ | 0.327 | 0.369 | 0.360 | 0.352 | 0.370 | 0.395 | 0.403 | 0.428 | 0.415 | 0.426 | 0.535 | 0.520 |
| ILI | 24 | _1.393_ | **0.768** | **1.319** | _0.754_ | 2.215 | 1.081 | 2.120 | 0.924 | 2.729 | 1.134 | 3.228 | 1.260 | 2.906 | 1.182 | 5.914 | 1.734 |
| | 36 | **1.456** | **0.794** | 1.579 | 0.870 | 1.963 | 0.963 | 1.955 | 0.917 | 2.451 | 1.001 | 2.679 | 1.080 | 2.585 | 1.038 | 6.631 | 1.845 |
| | 48 | **1.548** | **0.800** | _1.553_ | _0.815_ | 2.130 | 1.024 | 2.209 | 0.931 | 2.350 | 1.027 | 2.622 | 1.078 | 3.024 | 1.145 | 6.736 | 1.857 |
| | 60 | 1.498 | 0.797 | **1.470** | **0.788** | 2.368 | 1.096 | 2.001 | 0.925 | 2.518 | 1.056 | 2.847 | 1.144 | 2.761 | 1.114 | 6.870 | 1.879 |

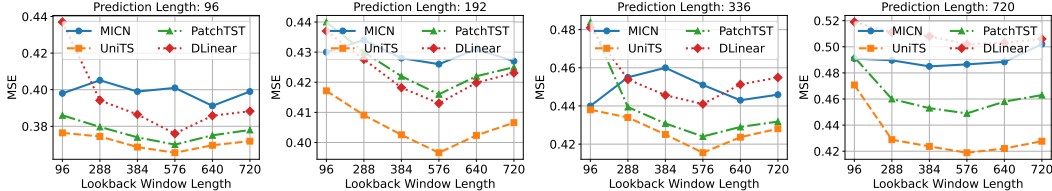

Figure 2: Multivariate long-term forecasting results are evaluated using prediction length $T \in \{96, 192, 336, 720\}$ on ETTh1 with lookback window length $L \in \{96, 288, 384, 576, 640, 720\}$.

## 4.2 PREDICTION PERFORMANCE WITH FINETUNED LOOKBACK LENGTH

Table 1 provides an overall view analysis of the prediction performance across all datasets with varying prediction lengths of all selected models with a finetuned lookback length, and our proposed UniTS demonstrates consistent superiority over all baseline methods. For instance, when focusing on the best-performing Transformer-based models, UniTS consistently achieves remarkable reductions in both MSE and MAE. Notably, in comparison to PatchTST, UniTS showcases a substantial reduction in MSE and in MAE across different dataset scenarios. Additionally, when compared to DLinear, UniTS consistently outperforms it, especially on datasets with complex patterns, such as Weather, Traffic, Electricity, and the ILI dataset. These findings underscore the effectiveness and robustness of UniTS as a forecasting model, making it a compelling choice for various multivariate time series forecasting tasks. Next, we will delve into the details of the model and validate the impact of each module on the model's predictive performance, allowing for a more in-depth interpretation of the outstanding performance displayed.

Table 2: Multivariate long-term forecasting results with different model ablations, evaluated using prediction length $T \in \{96, 192, 336, 720\}$ on Traffic, Electricity, and Weather. IN: Instance Normalization, LFE: Local Feature Extraction, GFE: Global Feature Extraction, PE: Position encoding, TE, Temporal Embedding, Attention: Attention Layer, LN: Layer Normalization.

| Models | | UniTS | | - IN | | - LFE | | - GFE | | + PE | | + TE | | + Attention | | + LN | |
|---|---|---|---|---|---|---|---|---|---|---|---|---|---|---|---|---|---|
| Metric | | MSE | MAE | MSE | MAE | MSE | MAE | MSE | MAE | MSE | MAE | MSE | MAE | MSE | MAE | MSE | MAE |
| Traffic | 96 | **0.355** | **0.243** | 0.390 | 0.261 | 0.365 | 0.254 | 0.488 | 0.320 | 0.360 | 0.245 | 0.357 | 0.244 | 0.370 | 0.259 | 0.358 | 0.244 |
| | 192 | **0.365** | **0.251** | 0.402 | 0.275 | 0.372 | 0.256 | 0.503 | 0.323 | 0.369 | 0.247 | 0.371 | 0.255 | 0.377 | 0.260 | 0.369 | 0.255 |
| | 336 | **0.380** | **0.260** | 0.412 | 0.283 | 0.388 | 0.263 | 0.518 | 0.329 | 0.383 | 0.263 | 0.384 | 0.264 | 0.394 | 0.265 | 0.384 | 0.263 |
| | 720 | **0.417** | **0.279** | 0.430 | 0.291 | 0.425 | 0.289 | 0.526 | 0.325 | 0.425 | 0.284 | 0.420 | 0.285 | 0.435 | 0.291 | 0.424 | 0.284 |
| Electricity | 96 | **0.130** | **0.221** | 0.141 | 0.237 | 0.132 | 0.223 | 0.188 | 0.290 | 0.133 | 0.225 | 0.132 | 0.223 | 0.135 | 0.228 | 0.133 | 0.224 |
| | 192 | **0.146** | **0.241** | 0.151 | 0.246 | 0.146 | 0.242 | 0.195 | 0.295 | 0.150 | 0.247 | 0.148 | 0.243 | 0.153 | 0.250 | 0.147 | 0.243 |
| | 336 | **0.159** | **0.256** | 0.165 | 0.263 | 0.167 | 0.265 | 0.207 | 0.312 | 0.161 | 0.258 | 0.161 | 0.257 | 0.164 | 0.261 | 0.161 | 0.260 |
| | 720 | **0.195** | **0.292** | 0.197 | 0.295 | 0.203 | 0.302 | 0.215 | 0.318 | 0.200 | 0.296 | 0.199 | 0.296 | 0.205 | 0.306 | 0.201 | 0.300 |
| Weather | 96 | **0.144** | **0.187** | 0.160 | 0.209 | 0.146 | 0.190 | 0.175 | 0.235 | 0.149 | 0.195 | 0.146 | 0.190 | 0.151 | 0.198 | 0.147 | 0.192 |
| | 192 | **0.187** | **0.240** | 0.211 | 0.259 | 0.188 | 0.242 | 0.233 | 0.270 | 0.192 | 0.244 | 0.190 | 0.242 | 0.193 | 0.247 | 0.192 | 0.246 |
| | 336 | **0.236** | **0.269** | 0.254 | 0.332 | 0.240 | 0.303 | 0.279 | 0.320 | 0.242 | 0.276 | 0.240 | 0.274 | 0.245 | 0.277 | 0.240 | 0.272 |
| | 720 | **0.303** | **0.318** | 0.315 | 0.342 | 0.313 | 0.357 | 0.351 | 0.394 | 0.309 | 0.328 | 0.305 | 0.321 | 0.310 | 0.331 | 0.305 | 0.322 |

## 4.3 PREDICTION PERFORMANCE WITH SPECIFIED LOOKBACK LENGTH

Our experiments found the effect of lookback window length is rather evident and can not be ignored. Figure 2 illustrates an example on the results of setting (ii) on dataset ETTh2. We can see that the performance curve usually exhibits an upward trend with increasing lookback window length. However, this performance gain is not indefinite and gradually plateaus, and in some cases, declines. It highlights the critical role of lookback window length in shaping temporal modeling performance, as well as the challenges posed by prolonged sequences in maintaining model effectiveness. This phenomenon warrants a deeper understanding and analysis for long-term temporal modeling and generalization. Notably, prior research often sidesteps this issue. Representative works like Informer, Autoformer, and TimesNet frequently employ fixed lookback window length for performance evaluation Zhou et al. (2021); Wu et al. (2021; 2022). Conversely, works such as DLinear, PatchTST, and MICN finetune the lookback window length. This discrepancy in evaluation practices introduces a degree of unfairness, as the lookback window length directly impacts the model's ultimate performance. Due to space constraints, we have provided additional experimental results in Appendix A.5 regarding the performance of each model across various fixed lookback window sizes.

## 4.4 IS TRANSFORMER-STYLE MODELING NECESSARY FOR TIME SERIES FORECASTING?

The discussion on whether Transformer (Vaswani et al., 2017) modeling is effective under the scenarios of time series forecasting have been existed for a time. The experiments with DLinear indicate that remarkable predictive results can be achieved by combining only one linear layer, surpassing the previous Transformer models. However, in subsequent work, PatchTST once again utilizes a Transformer-based structure to model time-series sequences and achieves even better performance. In this paper, we will continue to investigate whether the Transformer's modeling approach genuinely contributes to temporal modeling for time-series prediction. It's worth noting that Recent work Li et al. (2023) has suggested that, while PatchTST indeed outperforms the vanilla DLinear, this gain may not necessarily come from the Transformer's modeling approach but rather from their instance normalization (IN) method. It shows that removing IN from PatchTST even leads to performance lower than that of DLinear, and adding IN to DLinear makes its performance comparable to PatchTST. In this paper, we employ ablation experiments to more directly assess which of the two, Transformer and Linear networks, has the advantage in temporal modeling for time series forecasting. That is, if we further extend the linear layer by incorporating mechanisms used Transformer layers: attention layers, position encoding, and layer normalization, the resulting structure evolves into a Transformer-like architecture. We briefly elaborate on this transformation by introducing these additive modules:

**Attention Layer Addition**: Incorporating an attention mechanism introduces the capability for the model to focus on different parts of the input sequence while making predictions. This is achieved by calculating attention scores between different elements in the sequence. The attention operation can be represented as follows: Given input sequence of series $m$ after patching $\mathbf{X}_m^P \in \mathbb{R}^{N \times P}$, where

$N$ is the patch number and $P$ is the patch length, the attention mechanism computes a set of attention scores $A_i \in \mathbb{R}^{N \times N}$: $A_i = \text{Softmax}(Q_i K_i^T / \sqrt{d_k})$, $Q_i$, and $K_i$, are linear projections of $\mathbf{X}_m$ and embeddings of intermediate layers with hidden size $D$, and $d_k$ is the dimension of the key vectors.

**Position Encoding Addition**: To account for the positional information of elements in the sequence, position embeddings are added. These embeddings provide the model with knowledge of the order or position of elements within the sequence. The position embedding $PE_i \in \mathbb{R}^{L \times D}$ is added element-wise to the input sequence $h_i$: $h_{i_{\text{with PE}}} = h_i + PE_i$. The position embeddings are often pre-defined based on the position of elements in the sequence, details of position coding method can be found at Appendix A.2.

**Layer normalization Addition**: It is applied to stabilize and standardize the activations within each layer of the model. This helps improve training convergence and overall model performance. The layer normalization can be represented as: $h_i = \text{LayerNorm}(h_{i_{\text{with PE}}})$. $h_i$ represents the layer-normalized version of the input sequence with position embeddings.

Given that UniTS employs a hybrid modeling approach, we can further test the impact of above modules by adding it to the vanilla linear global feature extractor. Table 2 presents our ablation results on the Traffic, Electricity, and Weather datasets. It can be observed that, in all cases, the model's performance deteriorates after adding attention, which shows that the attention layer is not essential for temporal modeling in time-series forecasting models. Moreover, Instance Normalization (IN) significantly improves the model performance, indicating its crucial importance in temporal modeling. Position encoding and temporal embedding is widely used in sequence modeling within attention mechanisms (Nie et al., 2022; Das et al., 2023), designed to provide positional or temporal information to the model. However, our ablation experiments either revealed that in certain scenarios, position encoding does not necessarily contribute to improved predictive performance. This phenomenon can be attributed to the nature of the datasets. In cases where the inherent temporal order of the data is evident and well-captured by other model components, the addition of position encoding may introduce redundancy without a significant benefit. Furthermore, excessive reliance on position encoding can lead to overfitting when the temporal patterns are adequately captured by other model components. Based on our observations, we suggest that the inclusion of position encoding should be carefully considered, especially when the temporal relationships within the data are adequately captured by the model's architecture. A more judicious approach to position encoding can help streamline model complexity and maintain competitive predictive performance. Regarding layer normalization, we found through experiments that it does not directly benefit the model's performance. We believe this is because time-series models often have relatively shallow structures and fewer parameters, and they can achieve stable learning without the need for layer normalization.

### 4.5 MODEL ANALYSIS

We further verify the efficacy of the modules used in UniTS by conducting ablations and hyperparameter tuning. Our validation of the model modules primarily focuses on two aspects: (i) Whether the hybrid modeling approach can effectively enhance generalization across datasets; and (ii) How the impact of the hyperparamters like lookback window length $L$ on model performance.

**How Hybrid Modeling Benefits Time Series Forecasting**  Our experiments delved into the effectiveness of hybrid feature extraction methods within our UniTS framework. We abalte the Global Feature Extraction module (GFE) and Local Feature Extraction module (LFE) to evaluate the effectiveness of hybrid modeling. From the results in Table 2, we can observe that removing both feature extraction modules leads to a decrease in model performance. Removing LFE results in an average increase of 2.19% in MSE, while removing GFE leads to an average increase of 28.04% in MSE. This indicates that the hybrid temporal modeling approach, which combines both modules, is more advantageous compared to a single modeling approach. Additionally, it highlights the significant importance of GFE for the model's predictions. Our findings underscore the significance of incorporating hybrid feature extractors in time series modeling, and showing the importance of considering multi-scale feature extraction when designing predictive models. The results revealed that combining global and local feature extraction strategies through hybrid modeling consistently improved predictive performance across various datasets. This performance enhancement can be attributed to the critical role of hybrid feature extraction in capturing intricate temporal patterns.

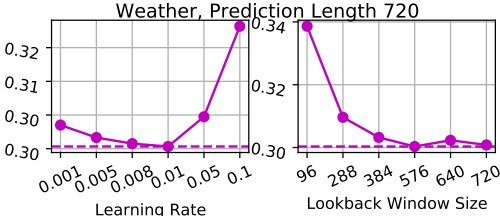

Figure 3: Results with different hyperparamter selections with learning rate and lookback window length, the y axis is MSE, a lower MSE indicates a better prediction.

**Good Hyperparameters Make a Good Neural Forecaster** Numerous studies have proposed various neural network architectures for multivariate time series forecasting. As we dicussed above, one prevailing issue lie in these works is the lack of standardized parameter design, leading to potential unfair comparisons and evaluations. Take the DLinear model for example. (Li et al., 2023) shows that DLinear can achieve comparable or even superior performance to PatchTST by adopting similar normalization strategies (ReVIN) (Kim et al., 2021), akin to those used in the PatchTST framework. Our experiments have shed light on this aspect.

We demonstrate that carefully chosen parameters is of critical importance for a good neural forecasting model. In addition, Figure 3 provides a simple example of how the learning rate and lookback window length have a

| | Traffic | | Weather | | ETTh1 | |
|---|---|---|---|---|---|---|
| $L$ | 96 | 720 | 96 | 720 | 96 | 720 |
| BO | 0.368 | 0.434 | 0.151 | 0.310 | 0.375 | 0.430 |
| Random | 0.385 | 0.485 | 0.160 | 0.330 | 0.397 | 0.475 |
| Oracle | **0.355** | **0.417** | **0.144** | **0.303** | **0.365** | **0.418** |

Table 3: Hyperparameter search results with different search strategies.

profound impact the UniTS's forecasting performance. It is evident that the choice of these two parameters plays a crucial role in determining the final performance of the model. To further evaluate the importance of hyperparameter selection in time series models. We conducted a comprehensive parameter exploration experiments on hyperpameters including learning rate, the use of lookback window length, learning rate, and hidden size, *e.t.c.*. Due to limited computational resources, we predefined a parameter space for UniTS, and the results can be found at Appendix A.6.

To further evaluate the important of parameter search of time series forecasting model, we employed three different parameter search methods: (1) Random: Using randomly selected parameter settings as the final model results; (2) BO: Bayesian Optimization-based parameter search method (Snoek et al., 2012); (3) Oracle: Utilizing Grid Search for the optimal results. Even though we manually constrained the parameter search space, we were able to keep (3) within the computational budget range available to us, despite the exponential increase in search difficulty with the number of parameters. Table 3 presents the results of the parameter search experiments. For each prediction task, the random search is the average of 20 search runs, while Bayesian search consists of 20 search iterations. It can be observed that compared to the random strategy, the Bayesian search strategy yields parameters that, on average, reduce the MSE by 7.51%. However, there is still a noticeable gap in performance compared to the parameters obtained through grid search for achieving optimal performance. This underscores the significance of designing efficient parameter search capabilities for time-series prediction models, which remains an area worth investigating. In all, we explored the utility of these modules and clarified how their selection impacts the overall model. Thus, this brings attention to the complexity introduced by parameter selection, a frequently overlooked issue in existing literature. Notably, in prior studies, model performance often resulted from manual selection within a given architecture. We delve into the intricacies of parameter selection and training costs in Appendix A.4, demonstrating that UniTS achieves a stable and effective prediction model with a limited computational budget.

## 5 CONCLUSION

In summary, our work contributes to the advancement of time series modeling with comprehensive experiments and offers a powerful and flexible framework for researchers and practitioners in the field. Our exploration has shed light on the critical roles of individual modules within these frameworks, uncovering valuable insights for the design of effective temporal prediction models. By decoupling of modules in previous works and a more refined performance analysis, we also have introduced a versatile and simple machine learning framework UniTS for temporal modeling for time series forecasting, bridging the gap between various learning paradigms. Through rigorous experimentats, we have demonstrated the superior performance of our proposed across diverse datasets. We provide comprehensive ablation experiments that enriches our understanding of the intricate relationship between model components.

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

## A    APPENDIX

### A.1    DATASET STATISTICS

In this study, we use 8 widely-used multivariate datasets for forecasting performance evaluation. We provide an overview of the experimental datasets as follows:

1. ETT[1] dataset comprises four sub-datasets: ETTh1, ETTh2, ETTm1, and ETTm2, collected from electricity transformers at intervals of 1 hour and 15 minutes, respectively, spanning from July 2016 to July 2018.

---

[1]https://github.com/zhouhaoyi/ETDataset

2. The Electricity[2] dataset includes hourly records of electricity consumption from 321 customers, spanning from 2012 to 2014.

3. Traffic[3] dataset contains data from the California Department of Transportation, providing hourly road occupancy rates measured by various sensors on San Francisco Bay area freeways.

4. Weather[4] dataset consists of 21 meteorological indicators, recorded at 10-minute intervals throughout the entire year of 2020.

5. ILI[5] dataset records weekly data on influenza-like illness (ILI) patients from the Centers for Disease Control and Prevention of the United States, covering the period from 2002 to 2021.

Table 4 summarizes the details of the used datasets

| Dataset | Length | Series Number | Record Frequency |
|---|---|---|---|
| ETTh | 17420 | 8 | 1 hour |
| ETTm | 69680 | 8 | 15 minutes |
| Electricity | 26304 | 322 | 1 hour |
| Traffic | 17544 | 863 | 1 hour |
| Weather | 52696 | 22 | 10 minutes |
| ILI | 966 | 8 | 7 days |

Table 4: Dataset Statistics.

## A.2 MODELS, HYPERPARAMETERS, AND REPRODUCIBILITY

We provide detailed description of the experimental setup for a easy reproducibility in this section.

### A.2.1 IMPLEMENTATION DETAILS ON UNITS

Our approach is trained using the L2 loss and employs the ADAM optimizer. The batch size is configured to be 32 for Traffic and Electricity datasets and 16 for others. The number of decompose module is set to 1, and the layer number of global feature extractor and local feature extractor is 1. The patch length and stride is set to 16 and 4 for ETT datasets and weather dataset. For Traffic and Electricity dataset, it's set to 32 and 8. We employ early stopping during training, which terminates the process after 5 epochs if there is no improvement in loss on the validation set. We evaluate our model using mean square error (MSE) and mean absolute error (MAE) as metrics. All experiments are conducted 3 times with distinct random seeds, implemented in PyTorch. All the experiments are conducted on 5 Azure virtual machines each with 4 NVIDIA RTX V100 16GB GPU.

**Position Encoding**  Sequence Position Embedding (PE) is a fundamental component in many sequence-to-sequence models, including Transformer-based architectures. The basic idea behind Sequence Position Embedding is to add position-specific information to the input data, allowing the model to discern not only the content but also the location of each element in the sequence.

Mathematically, Sequence Position Embedding is commonly represented as follows:

$$PE_{(pos,2i)} = \sin\left(\frac{pos}{10000^{2i/d}}\right)$$

$$PE_{(pos,2i+1)} = \cos\left(\frac{pos}{10000^{2i/d}}\right)$$

Here, $PE_{(pos,2i)}$ and $PE_{(pos,2i+1)}$ represent the position embeddings for a given position $pos$ and dimension $2i$ and $2i + 1$ respectively. The function $\sin$ and $\cos$ introduce oscillatory patterns that are designed to capture different positional information.

---

[2]https://archive.ics.uci.edu/ml/datasets/ElectricityLoadDiagrams20112014
[3]https://pems.dot.ca.gov/
[4]https://www.bgc-jena.mpg.de/wetter/
[5]https://gis.cdc.gov/grasp/fluview/fluportaldashboard.html

These position embeddings are added to the input sequence elements element-wise, enriching the input representation with positional context. This allows the model to take into account the sequence order during its computations, which is particularly valuable for tasks where the order of elements carries significance, such as language understanding and time series forecasting.

### A.2.2 IMPLEMENTATION DETAILS ON BASELINE MODELS

In the experimental section of this paper, for the selection of the optimal lookback window size, we referred to the reported results from PatchTST (Nie et al., 2022). They conducted fine-tuning experiments for a given $L \in \{24, 48, 96, 192, 336, 720\}$. However, since the original experiments in TimesNet and MICN used a fixed input sequence length of 96 for predictions, we re-ran these two models in this experimental setting.

For the experiments with a fixed lookback window size, we utilized the results from TimesNet (Wu et al., 2022) and MICN (Wang et al., 2022). However, since PatchTST and DLinear did not provide experimental results with the same lookback window size as TimesNet ($L = 96$), and as shown in Figure 2, we aimed to compare several groups of experiments with gradually increasing lookback window size, $L \in \{96, 288, 384, 576, 720\}$. We conducted experiments for TimesNet, MICN, PatchTST, DLinear, and UniTS in this context. To ensure a fair comparison, all baseline models and UniTS were trained using the optimal hyperparameters found through grid search for the fixed lookback window size scenario.

Regarding TimesNet (Wu et al., 2022), due to its frequent Fourier transform calculations and the use of serial computation in its implementation, the extensive time overhead made it exceedingly difficult to explore the desired lookback window sizes comprehensively. One feasible solution is to reduce the hyperparamter search space for TimesNet. However, reducing the search space would result in an unfair comparison, leading us to conclude that reliable TimesNet results cannot be provided in this setting. Consequently, its performance in this context is not mentioned in this paper. We used the original official code repository of TimesNet[6], DLinear[7], PatchTST[8], and MICN[9], to generate corresponding experimental results.

It is worth noting that, recently, TiDE was proposed as a linear temporal prediction model that combines temporal embedding and encoder-decoder structures (Das et al., 2023). However, as of the writing of this work, the official code for this work has not been released. Based on the original paper's description, we attempted to implement this method but did not successfully reproduce the results as described in the work. Out of caution, we did not include the experimental results of this method in our discussions. Information regarding our implementation of this method and its results can be found in our provided code repository at `https://anonymous.4open.science/r/UniTS-8DA8/README.md`.

### A.3 FURTHER EXPERIMENTAL RESULTS WITH ADDITIONAL BASELINES

We provide further experimental results compared with baselines, including RMLP and RLinear (Li et al., 2023). RMLP corresponds to a single-layer linear network utilizing ReVIN, while RLinear employs a two-layer linear network with ReVIN. Additionally, we compare against N-BEATS (Oreshkin et al., 2019) and N-HiTS (Challu et al., 2023), both utilizing a scalable linear structure with a residual architecture. The SpaceTimeFormer (STF) (Grigsby et al., 2021) is a long-term time series prediction model incorporating context embedding and attention structures. For N-BEATS, N-HiTS, and SpaceTimeFormer, we conducted a search over network depths ranging from 1 to 4 layers. The hidden size was explored within the range of 16, 32, and 64. The final results can be seen from Table 5. We can observe that both RMLP and RLinear achieve excellent performance, consistent with the observations in this study: the use of the ReVIN (Kim et al., 2021) method effectively enhances the predictive performance of the model on the dataset employed in this paper. In contrast, the performance of N-HiTS and N-BEATS, which do not utilize ReVIN, is relatively poorer. Additionally, SpaceTimeFormer, employing attention and context embedding, exhibits slightly infe-

---

[6] `https://github.com/thuml/TimesNet`

[7] `https://github.com/vivva/DLinear`

[8] `https://github.com/yuqinie98/PatchTST/`

[9] `https://github.com/wanghq21/MICN/`

Table 5: Multivariate long-term forecasting results evaluated using prediction length $T \in \{96, 192, 336, 720\}$. A lower MSE or MAE indicates a better prediction. The best results are highlighted in **bold**.

| Models | | UniTS | | RMLP | | RLinear | | N-BEATS | | N-HiTS | | TiDE | | STF | |
|---|---|---|---|---|---|---|---|---|---|---|---|---|---|---|---|---|
| Metric | | MSE | MAE | MSE | MAE | MSE | MAE | MSE | MAE | MSE | MAE | MSE | MAE | MSE | MAE |
| ETTh1 | 96 | **0.365** | **0.390** | 0.368 | 0.395 | **0.365** | 0.391 | 0.384 | 0.400 | 0.391 | 0.420 | 0.376 | 0.398 | 0.387 | 0.405 |
| | 192 | **0.396** | **0.411** | 0.405 | 0.417 | 0.412 | 0.420 | 0.430 | 0.445 | 0.430 | 0.453 | 0.420 | 0.432 | 0.445 | 0.433 |
| | 336 | **0.414** | **0.426** | 0.425 | 0.439 | 0.439 | 0.443 | 0.479 | 0.460 | 0.439 | 0.458 | 0.437 | 0.456 | 0.465 | 0.447 |
| | 720 | **0.418** | **0.435** | 0.430 | 0.448 | 0.501 | 0.490 | 0.513 | 0.489 | 0.485 | 0.502 | 0.450 | 0.461 | 0.488 | 0.508 |
| ETTh2 | 96 | **0.262** | **0.329** | 0.270 | 0.335 | 0.263 | 0.331 | 0.335 | 0.369 | 0.331 | 0.375 | 0.278 | 0.340 | 0.290 | 0.349 |
| | 192 | **0.327** | **0.360** | 0.339 | 0.379 | 0.330 | 0.362 | 0.397 | 0.405 | 0.419 | 0.437 | 0.350 | 0.389 | 0.365 | 0.399 |
| | 336 | **0.349** | **0.384** | 0.363 | 0.393 | 0.358 | 0.385 | 0.449 | 0.450 | 0.442 | 0.468 | 0.375 | 0.417 | 0.382 | 0.426 |
| | 720 | **0.377** | **0.415** | 0.379 | 0.422 | 0.382 | 0.423 | 0.451 | 0.455 | 0.439 | 0.462 | 0.385 | 0.430 | 0.398 | 0.408 |
| ETTm1 | 96 | **0.289** | **0.343** | 0.299 | 0.347 | 0.301 | 0.349 | 0.330 | 0.365 | 0.344 | 0.372 | 0.307 | 0.351 | 0.318 | 0.357 |
| | 192 | **0.326** | **0.360** | 0.333 | 0.362 | 0.334 | 0.364 | 0.370 | 0.382 | 0.375 | 0.388 | 0.342 | 0.365 | 0.347 | 0.374 |
| | 336 | **0.355** | **0.382** | 0.369 | 0.388 | 0.370 | 0.390 | 0.402 | 0.400 | 0.401 | 0.404 | 0.378 | 0.396 | 0.409 | 0.411 |
| | 720 | **0.406** | **0.408** | 0.416 | 0.415 | 0.420 | 0.418 | 0.453 | 0.461 | 0.447 | 0.458 | 0.433 | 0.439 | 0.461 | 0.467 |
| ETTm2 | 96 | **0.159** | **0.250** | 0.164 | 0.253 | 0.161 | 0.252 | 0.180 | 0.258 | 0.185 | 0.264 | 0.168 | 0.256 | 0.175 | 0.265 |
| | 192 | **0.213** | **0.285** | 0.221 | 0.290 | 0.215 | 0.287 | 0.244 | 0.302 | 0.251 | 0.309 | 0.225 | 0.293 | 0.231 | 0.296 |
| | 336 | **0.264** | **0.311** | 0.281 | 0.323 | 0.273 | 0.319 | 0.311 | 0.339 | 0.306 | 0.338 | 0.287 | 0.328 | 0.295 | 0.233 |
| | 720 | **0.344** | **0.353** | 0.369 | 0.371 | 0.362 | 0.366 | 0.389 | 0.401 | 0.385 | 0.400 | 0.375 | 0.378 | 0.380 | 0.402 |
| Traffic | 96 | **0.355** | **0.243** | 0.367 | 0.254 | 0.369 | 0.256 | 0.373 | 0.258 | 0.375 | 0.259 | 0.373 | 0.260 | 0.378 | 0.265 |
| | 192 | **0.365** | **0.251** | 0.377 | 0.258 | 0.382 | 0.265 | 0.409 | 0.277 | 0.415 | 0.283 | 0.406 | 0.280 | 0.418 | 0.285 |
| | 336 | **0.380** | **0.260** | 0.391 | 0.269 | 0.398 | 0.278 | 0.425 | 0.287 | 0.419 | 0.290 | 0.430 | 0.296 | 0.443 | 0.301 |
| | 720 | **0.417** | **0.279** | 0.432 | 0.287 | 0.441 | 0.300 | 0.468 | 0.312 | 0.472 | 0.316 | 0.447 | 0.303 | 0.465 | 0.310 |
| Electricity | 96 | **0.130** | **0.221** | 0.139 | 0.233 | 0.140 | 0.234 | 0.167 | 0.271 | 0.171 | 0.274 | 0.143 | 0.236 | 0.146 | 0.242 |
| | 192 | **0.146** | **0.241** | 0.150 | 0.247 | 0.149 | 0.249 | 0.184 | 0.285 | 0.189 | 0.289 | 0.152 | 0.250 | 0.157 | 0.254 |
| | 336 | **0.159** | **0.256** | 0.161 | 0.260 | 0.168 | 0.269 | 0.197 | 0.298 | 0.205 | 0.301 | 0.170 | 0.273 | 0.179 | 0.267 |
| | 720 | **0.195** | **0.292** | 0.199 | 0.298 | 0.202 | 0.304 | 0.218 | 0.315 | 0.221 | 0.326 | 0.205 | 0.309 | 0.213 | 0.304 |
| Weather | 96 | **0.144** | **0.189** | 0.149 | 0.197 | 0.162 | 0.211 | 0.172 | 0.219 | 0.182 | 0.233 | 0.170 | 0.217 | 0.185 | 0.237 |
| | 192 | **0.187** | **0.240** | 0.193 | 0.246 | 0.203 | 0.252 | 0.217 | 0.261 | 0.223 | 0.265 | 0.209 | 0.256 | 0.226 | 0.267 |
| | 336 | **0.236** | **0.271** | 0.240 | 0.273 | 0.251 | 0.287 | 0.277 | 0.304 | 0.281 | 0.332 | 0.258 | 0.294 | 0.284 | 0.275 |
| | 720 | **0.303** | **0.320** | 0.314 | 0.334 | 0.323 | 0.342 | 0.356 | 0.370 | 0.360 | 0.375 | 0.332 | 0.352 | 0.358 | 0.376 |
| ILI | 24 | **1.393** | **0.768** | 1.678 | 0.812 | 1.891 | 0.855 | 2.012 | 0.933 | 2.168 | 0.956 | 2.087 | 0.905 | 1.796 | 0.947 |
| | 36 | **1.456** | **0.794** | 1.620 | 0.831 | 1.523 | 0.873 | 1.955 | 0.876 | 1.980 | 0.871 | 1.678 | 0.870 | 2.215 | 0.978 |
| | 48 | **1.548** | **0.800** | 1.957 | 0.885 | 1.678 | 0.831 | 1.931 | 0.918 | 2.344 | 0.959 | 1.802 | 0.883 | 2.058 | 1.012 |
| | 60 | **1.498** | **0.797** | 1.855 | 0.864 | 1.776 | 0.894 | 2.332 | 0.943 | 2.071 | 0.976 | 1.756 | 0.891 | 2.199 | 0.930 |

Table 6: Training Cost *v.s.* Params.

| Model | UniTS | PatchTST | DLinear | TimesNet | MICN | Autoformer |
|---|---|---|---|---|---|---|
| Training Cost/Epoch (Seconds) | 3.908 | 6.972 | **2.790** | 10.135 | 7.72 | 19.58 |
| Params (Million) | 0.209 | 1.124 | **0.021** | 4.831 | 1.127 | 4.715 |

rior performance compared to other methods, aligning with other experimental observations in this paper: the use of attention and context embedding can impact model performance.

## A.4 COMPLEXITY ANALYSIS

The discussions within our paper introduce a novel method aimed at disentangling previously interconnected modules within various temporal models. We delve into the utility of these modules and elucidate how their selection impacts the overall model. This thrusts parameter selection, often overlooked in existing literature, into the spotlight as an unavoidable concern. Notably, the performance of previously discussed models has frequently relied on manual selection within a given architecture, making it a pivotal focus in our paper.

Directly addressing this issue presents challenges on two fronts: first, the exhaustive exploration of all possibilities for each model is unfeasible; second, a parameter search in a predefined space may introduce human bias, potentially leading to unfair comparisons. Despite these challenges, based on real-world dataset performance, we propose a comparative method within limited scenarios. In practical terms, the most effective time series prediction models tend to avoid overly complex network structures or high-dimensional parameters. Through our experiments, we have observed a negative impact on performance when structures are overly complex and parameterized.

From the perspective of the dataset used and the models employed in this paper, if our goal is to find an optimal model, we can define a search space. In actual experiments, we found that for almost all models, we only need to search within a relatively small hidden size and a shallow network range. Therefore, for all models, the number of times they select optimal network parameters is relatively

Table 7: Multivariate long-term forecasting results evaluated using a fixed lookback horizon ($L = 96$).

| Models | | UniTS | | PatchTST | | DLinear | | TimesNet | | MICN | | Fedformer | | Autoformer | |
|---|---|---|---|---|---|---|---|---|---|---|---|---|---|---|---|
| Metric | | MSE | MAE | MSE | MAE | MSE | MAE | MSE | MAE | MSE | MAE | MSE | MAE | MSE | MAE |
| Traffic | 96 | **0.574** | **0.341** | 0.590 | 0.367 | 0.650 | 0.396 | 0.593 | 0.321 | 0.575 | 0.344 | 0.587 | 0.366 | 0.597 | 0.371 |
| | 192 | **0.557** | **0.334** | 0.595 | 0.368 | 0.598 | 0.370 | 0.617 | 0.336 | 0.580 | 0.349 | 0.604 | 0.373 | 0.607 | 0.382 |
| | 336 | **0.566** | **0.331** | 0.603 | 0.370 | 0.605 | 0.373 | 0.629 | 0.336 | 0.583 | 0.345 | 0.621 | 0.383 | 0.623 | 0.387 |
| | 720 | **0.582** | **0.348** | 0.599 | 0.368 | 0.645 | 0.394 | 0.640 | 0.350 | 0.601 | 0.363 | 0.626 | 0.382 | 0.639 | 0.395 |
| Weather | 96 | **0.168** | **0.219** | 0.174 | 0.221 | 0.196 | 0.255 | 0.172 | 0.220 | 0.183 | 0.250 | 0.217 | 0.296 | 0.249 | 0.329 |
| | 192 | **0.210** | **0.259** | 0.222 | 0.264 | 0.237 | 0.296 | 0.219 | 0.261 | 0.246 | 0.317 | 0.276 | 0.336 | 0.325 | 0.370 |
| | 336 | **0.274** | **0.321** | 0.277 | 0.339 | 0.283 | 0.335 | 0.280 | 0.306 | 0.293 | 0.335 | 0.339 | 0.380 | 0.351 | 0.391 |
| | 720 | **0.348** | **0.385** | 0.361 | 0.367 | 0.403 | 0.428 | 0.365 | 0.359 | 0.373 | 0.399 | 0.403 | 0.428 | 0.415 | 0.426 |
| Electricity | 96 | **0.160** | **0.255** | 0.165 | 0.265 | 0.197 | 0.282 | 0.168 | 0.272 | 0.193 | 0.308 | 0.193 | 0.308 | 0.201 | 0.317 |
| | 192 | **0.180** | **0.278** | 0.186 | 0.291 | 0.196 | 0.285 | 0.184 | 0.289 | 0.200 | 0.308 | 0.201 | 0.315 | 0.222 | 0.334 |
| | 336 | **0.194** | **0.281** | 0.202 | 0.300 | 0.209 | 0.301 | 0.198 | 0.300 | 0.219 | 0.328 | 0.214 | 0.329 | 0.231 | 0.338 |
| | 720 | **0.210** | **0.304** | 0.215 | 0.317 | 0.245 | 0.333 | 0.220 | 0.320 | 0.224 | 0.332 | 0.246 | 0.355 | 0.254 | 0.361 |
| ETTh1 | 96 | **0.376** | **0.390** | 0.386 | 0.402 | 0.386 | 0.400 | 0.384 | 0.402 | 0.398 | 0.427 | 0.376 | 0.419 | 0.449 | 0.459 |
| | 192 | **0.417** | **0.445** | 0.440 | 0.438 | 0.437 | 0.432 | 0.436 | 0.429 | 0.430 | 0.453 | 0.420 | 0.448 | 0.500 | 0.482 |
| | 336 | 0.440 | 0.462 | 0.484 | 0.462 | 0.481 | 0.459 | 0.491 | 0.469 | **0.439** | **0.460** | 0.459 | 0.465 | 0.521 | 0.496 |
| | 720 | **0.470** | **0.480** | 0.492 | 0.497 | 0.519 | 0.516 | 0.521 | 0.500 | 0.491 | 0.509 | 0.506 | 0.507 | 0.514 | 0.512 |
| ETTh2 | 96 | **0.284** | **0.350** | 0.285 | 0.352 | 0.333 | 0.387 | 0.340 | 0.374 | 0.332 | 0.377 | 0.358 | 0.397 | 0.346 | 0.388 |
| | 192 | **0.368** | **0.402** | 0.369 | 0.404 | 0.477 | 0.476 | 0.402 | 0.414 | 0.422 | 0.441 | 0.429 | 0.439 | 0.456 | 0.452 |
| | 336 | **0.411** | **0.465** | 0.412 | 0.468 | 0.594 | 0.541 | 0.452 | 0.452 | 0.447 | 0.474 | 0.496 | 0.487 | 0.482 | 0.486 |
| | 720 | **0.415** | **0.479** | 0.421 | 0.485 | 0.831 | 0.657 | 0.462 | 0.468 | 0.442 | 0.467 | 0.463 | 0.474 | 0.515 | 0.511 |
| ETTm1 | 96 | **0.323** | **0.355** | 0.328 | 0.361 | 0.345 | 0.372 | 0.338 | 0.375 | 0.360 | 0.399 | 0.379 | 0.419 | 0.505 | 0.475 |
| | 192 | **0.368** | **0.402** | **0.368** | 0.403 | 0.380 | 0.389 | 0.374 | 0.387 | 0.402 | 0.426 | 0.426 | 0.441 | 0.553 | 0.496 |
| | 336 | **0.400** | **0.381** | 0.411 | 0.465 | 0.413 | 0.413 | 0.410 | 0.411 | 0.403 | 0.437 | 0.445 | 0.459 | 0.621 | 0.537 |
| | 720 | **0.415** | **0.416** | 0.467 | 0.550 | 0.474 | 0.453 | 0.478 | 0.450 | 0.459 | 0.464 | 0.543 | 0.490 | 0.671 | 0.516 |
| ETTm2 | 96 | **0.172** | **0.231** | 0.178 | 0.246 | 0.193 | 0.292 | 0.187 | 0.267 | 0.203 | 0.287 | 0.203 | 0.287 | 0.255 | 0.339 |
| | 192 | **0.232** | **0.279** | 0.245 | 0.302 | 0.284 | 0.362 | 0.249 | 0.309 | 0.262 | 0.326 | 0.269 | 0.328 | 0.281 | 0.340 |
| | 336 | **0.297** | **0.373** | 0.304 | 0.350 | 0.369 | 0.427 | 0.321 | 0.351 | 0.305 | 0.353 | 0.325 | 0.366 | 0.339 | 0.372 |
| | 720 | **0.391** | **0.408** | 0.403 | 0.415 | 0.554 | 0.522 | 0.408 | 0.403 | 0.389 | 0.407 | 0.421 | 0.415 | 0.433 | 0.432 |

small. From this perspective, in cases where the search space difference is not significant, in terms of computational complexity, the cost of parameter search mainly comes from the training cost of each parameter configuration, which can be further quantified by the training cost per epoch for each model. Therefore, based on this, we provide an example of the per epoch training cost for each model on the ETTh1 dataset in Table 6. In addition, to more comprehensively evaluate the spatial complexity of each model, we also provide the number of model parameters searched on the ETTh1 dataset. It can be observed that due to its simple and effective performance, DLinear exhibits the highest efficiency in terms of spatial complexity and inference time. In contrast, models using attention mechanisms typically require more parameters and longer inference times. In comparison, our proposed UniTS model maintains excellent performance while ensuring stable and efficient training efficiency and relatively low spatial occupation.

It is essential to acknowledge that the provided comparison is not flawless and does not claim absolute fairness. Nevertheless, we have demonstrated that this search method is simple and feasible within our existing knowledge. Within a limited time cost, using the proposed structure, we can train models that consistently achieve effective time series prediction performance. Furthermore, we would like to emphasize that certain existing works, by coupling numerous modules, have overlooked discussions on the search space. This oversight renders their comparisons highly unfair, as some models may adopt longer lookback window sizes, introducing bias into the comparison of effects. In our paper, when comparing the performance of the proposed models, we have designed experiments not only to address this gap in the comparison approach but also to explore the potential of further designing a search method within an effective parameter space. This aims to construct a more robust and effective temporal prediction model by integrating existing time series prediction modules.

## A.5 FULL FORECASTING PERFORMANCE WITH FIXED LOOKBACK WINDOW LENGTHS

We provide a full forecasting performance with lookback window size, $L = 96$, $L = 384$, and $L = 720$ for UniTS, PatchTST, DLinear, Timesnet, MICN, Fedformer and Autoformer, which can be seen through Table 7, Table 8, and Table 9.

Table 8: Multivariate long-term forecasting results evaluated using a fixed lookback horizon ($L = 384$).

| Models | | UniTS | | PatchTST | | DLinear | | TimesNet | | MICN | | Fedformer | | Autoformer | |
|---|---|---|---|---|---|---|---|---|---|---|---|---|---|---|---|
| Metric | | MSE | MAE | MSE | MAE | MSE | MAE | MSE | MAE | MSE | MAE | MSE | MAE | MSE | MAE |
| Traffic | 96 | **0.374** | **0.257** | 0.387 | 0.259 | 0.395 | 0.269 | 0.587 | 0.365 | 0.520 | 0.331 | 0.591 | 0.367 | 0.601 | 0.368 |
| | 192 | **0.388** | **0.263** | 0.398 | 0.268 | 0.404 | 0.267 | 0.609 | 0.372 | 0.547 | 0.339 | 0.608 | 0.371 | 0.609 | 0.370 |
| | 336 | **0.410** | **0.281** | 0.421 | 0.285 | 0.425 | 0.284 | 0.618 | 0.373 | 0.530 | 0.335 | 0.627 | 0.378 | 0.627 | 0.375 |
| | 720 | **0.429** | **0.287** | 0.438 | 0.295 | 0.446 | 0.303 | 0.621 | 0.373 | 0.571 | 0.352 | 0.645 | 0.383 | 0.641 | 0.386 |
| Weather | 96 | **0.145** | **0.190** | 0.151 | 0.203 | 0.154 | 0.208 | 0.182 | 0.247 | 0.185 | 0.251 | 0.217 | 0.265 | 0.251 | 0.305 |
| | 192 | **0.188** | **0.240** | 0.194 | 0.247 | 0.204 | 0.263 | 0.237 | 0.275 | 0.251 | 0.297 | 0.280 | 0.313 | 0.326 | 0.361 |
| | 336 | **0.237** | **0.273** | 0.257 | 0.301 | 0.266 | 0.311 | 0.297 | 0.331 | 0.301 | 0.330 | 0.343 | 0.371 | 0.358 | 0.395 |
| | 720 | **0.305** | **0.334** | 0.324 | 0.358 | 0.338 | 0.370 | 0.361 | 0.382 | 0.372 | 0.397 | 0.409 | 0.423 | 0.421 | 0.428 |
| Electricity | 96 | **0.132** | **0.225** | 0.141 | 0.237 | 0.158 | 0.255 | 0.170 | 0.275 | 0.192 | 0.297 | 0.195 | 0.295 | 0.201 | 0.314 |
| | 192 | **0.149** | **0.245** | 0.162 | 0.268 | 0.173 | 0.279 | 0.203 | 0.286 | 0.203 | 0.306 | 0.204 | 0.302 | 0.223 | 0.327 |
| | 336 | **0.168** | **0.274** | 0.179 | 0.283 | 0.203 | 0.302 | 0.203 | 0.303 | 0.215 | 0.324 | 0.215 | 0.311 | 0.231 | 0.336 |
| | 720 | **0.204** | **0.302** | 0.210 | 0.310 | 0.214 | 0.315 | 0.227 | 0.330 | 0.223 | 0.329 | 0.246 | 0.353 | 0.256 | 0.367 |
| ETTh1 | 96 | **0.368** | **0.395** | 0.373 | 0.400 | 0.386 | 0.406 | 0.391 | 0.403 | 0.398 | 0.423 | 0.380 | 0.428 | 0.441 | 0.453 |
| | 192 | **0.402** | **0.418** | 0.409 | 0.430 | 0.418 | 0.430 | 0.423 | 0.449 | 0.435 | 0.450 | 0.423 | 0.454 | 0.458 | 0.460 |
| | 336 | **0.423** | **0.439** | 0.430 | 0.444 | 0.445 | 0.451 | 0.489 | 0.478 | 0.448 | 0.466 | 0.461 | 0.468 | 0.485 | 0.499 |
| | 720 | **0.425** | **0.442** | 0.450 | 0.455 | 0.508 | 0.492 | 0.523 | 0.502 | 0.490 | 0.508 | 0.510 | 0.515 | 0.523 | 0.525 |
| ETTh2 | 96 | **0.268** | **0.337** | 0.275 | 0.342 | 0.295 | 0.355 | 0.345 | 0.377 | 0.335 | 0.380 | 0.360 | 0.401 | 0.346 | 0.374 |
| | 192 | **0.328** | **0.363** | 0.340 | 0.383 | 0.388 | 0.428 | 0.410 | 0.413 | 0.435 | 0.451 | 0.435 | 0.451 | 0.443 | 0.447 |
| | 336 | **0.349** | **0.385** | 0.361 | 0.398 | 0.437 | 0.450 | 0.455 | 0.460 | 0.451 | 0.473 | 0.501 | 0.508 | 0.482 | 0.480 |
| | 720 | **0.380** | **0.420** | 0.386 | 0.427 | 0.470 | 0.485 | 0.470 | 0.464 | 0.443 | 0.469 | 0.478 | 0.485 | 0.468 | 0.502 |
| ETTm1 | 96 | **0.289** | **0.350** | 0.297 | **0.350** | 0.304 | 0.355 | 0.343 | 0.372 | 0.360 | 0.399 | 0.386 | 0.428 | 0.513 | 0.512 |
| | 192 | **0.328** | **0.367** | 0.341 | 0.369 | 0.337 | 0.368 | 0.381 | 0.395 | 0.413 | 0.418 | 0.430 | 0.443 | 0.517 | 0.520 |
| | 336 | **0.362** | **0.383** | 0.371 | 0.398 | 0.370 | 0.388 | 0.411 | 0.418 | 0.417 | 0.415 | 0.453 | 0.470 | 0.523 | 0.527 |
| | 720 | **0.410** | **0.415** | 0.425 | 0.433 | 0.424 | 0.429 | 0.483 | 0.468 | 0.460 | 0.458 | 0.557 | 0.501 | 0.548 | 0.537 |
| ETTm2 | 96 | **0.162** | **0.253** | 0.164 | 0.258 | 0.171 | 0.268 | 0.190 | 0.264 | 0.210 | 0.288 | 0.205 | 0.291 | 0.210 | 0.302 |
| | 192 | **0.216** | **0.290** | 0.225 | 0.303 | 0.233 | 0.306 | 0.250 | 0.305 | 0.273 | 0.327 | 0.276 | 0.343 | 0.285 | 0.346 |
| | 336 | **0.267** | **0.316** | 0.279 | 0.334 | 0.297 | 0.348 | 0.333 | 0.346 | 0.325 | 0.359 | 0.331 | 0.374 | 0.350 | 0.398 |
| | 720 | **0.353** | **0.360** | 0.362 | 0.388 | 0.390 | 0.416 | 0.419 | 0.410 | 0.399 | 0.407 | 0.427 | 0.420 | 0.416 | 0.426 |

Table 9: Multivariate long-term forecasting results evaluated using a fixed lookback horizon ($L = 720$).

| Models | | UniTS | | PatchTST | | DLinear | | TimesNet | | MICN | | Fedformer | | Autoformer | |
|---|---|---|---|---|---|---|---|---|---|---|---|---|---|---|---|
| Metric | | MSE | MAE | MSE | MAE | MSE | MAE | MSE | MAE | MSE | MAE | MSE | MAE | MSE | MAE |
| Traffic | 96 | **0.376** | **0.258** | 0.393 | 0.267 | 0.395 | 0.270 | 0.588 | 0.370 | 0.524 | 0.336 | 0.592 | 0.370 | 0.605 | 0.371 |
| | 192 | **0.390** | **0.263** | 0.408 | 0.275 | 0.412 | 0.278 | 0.611 | 0.343 | 0.550 | 0.343 | 0.610 | 0.375 | 0.609 | 0.377 |
| | 336 | **0.413** | **0.284** | 0.427 | 0.291 | 0.433 | 0.295 | 0.620 | 0.378 | 0.542 | 0.338 | 0.632 | 0.391 | 0.630 | 0.378 |
| | 720 | **0.431** | **0.260** | 0.442 | 0.299 | 0.450 | 0.304 | 0.624 | 0.377 | 0.575 | 0.358 | 0.649 | 0.386 | 0.635 | 0.391 |
| Weather | 96 | **0.148** | **0.193** | 0.157 | 0.209 | 0.162 | 0.211 | 0.186 | 0.249 | 0.191 | 0.258 | 0.220 | 0.271 | 0.256 | 0.307 |
| | 192 | **0.190** | **0.240** | 0.198 | 0.254 | 0.201 | 0.267 | 0.239 | 0.277 | 0.260 | 0.301 | 0.285 | 0.318 | 0.324 | 0.369 |
| | 336 | **0.240** | **0.276** | 0.265 | 0.309 | 0.268 | 0.315 | 0.300 | 0.334 | 0.305 | 0.332 | 0.349 | 0.377 | 0.361 | 0.402 |
| | 720 | **0.309** | **0.337** | 0.331 | 0.363 | 0.344 | 0.370 | 0.365 | 0.383 | 0.375 | 0.402 | 0.413 | 0.429 | 0.424 | 0.435 |
| Electricity | 96 | **0.135** | **0.229** | 0.148 | 0.242 | 0.161 | 0.259 | 0.173 | 0.282 | 0.192 | 0.300 | 0.196 | 0.300 | 0.199 | 0.322 |
| | 192 | **0.150** | **0.249** | 0.166 | 0.273 | 0.179 | 0.286 | 0.187 | 0.289 | 0.203 | 0.306 | 0.208 | 0.304 | 0.228 | 0.331 |
| | 336 | **0.170** | **0.280** | 0.186 | 0.290 | 0.207 | 0.308 | 0.206 | 0.306 | 0.215 | 0.324 | 0.223 | 0.315 | 0.235 | 0.342 |
| | 720 | **0.207** | **0.308** | 0.216 | 0.313 | 0.218 | 0.319 | 0.230 | 0.331 | 0.223 | 0.329 | 0.255 | 0.365 | 0.260 | 0.370 |
| ETTh1 | 96 | **0.371** | **0.401** | 0.380 | 0.412 | 0.391 | 0.411 | 0.395 | 0.408 | 0.398 | 0.423 | 0.384 | 0.433 | 0.447 | 0.459 |
| | 192 | **0.404** | **0.423** | 0.418 | 0.436 | 0.419 | 0.432 | 0.440 | 0.451 | 0.435 | 0.450 | 0.427 | 0.460 | 0.459 | 0.467 |
| | 336 | **0.424** | **0.441** | 0.437 | 0.448 | 0.451 | 0.456 | 0.494 | 0.479 | 0.448 | 0.466 | 0.465 | 0.471 | 0.492 | 0.503 |
| | 720 | **0.428** | **0.445** | 0.455 | 0.461 | 0.513 | 0.498 | 0.528 | 0.510 | 0.490 | 0.508 | 0.513 | 0.519 | 0.528 | 0.528 |
| ETTh2 | 96 | **0.274** | **0.341** | 0.282 | 0.345 | 0.297 | 0.360 | 0.348 | 0.380 | 0.335 | 0.380 | 0.368 | 0.410 | 0.348 | 0.376 |
| | 192 | **0.333** | **0.371** | 0.346 | 0.389 | 0.391 | 0.432 | 0.414 | 0.417 | 0.423 | 0.438 | 0.442 | 0.457 | 0.449 | 0.452 |
| | 336 | **0.353** | **0.393** | 0.370 | 0.405 | 0.443 | 0.455 | 0.456 | 0.461 | 0.451 | 0.473 | 0.505 | 0.510 | 0.487 | 0.489 |
| | 720 | **0.383** | **0.428** | 0.391 | 0.428 | 0.472 | 0.488 | 0.477 | 0.470 | 0.443 | 0.469 | 0.478 | 0.486 | 0.473 | 0.505 |
| ETTm1 | 96 | **0.294** | **0.360** | 0.302 | 0.367 | 0.305 | 0.357 | 0.351 | 0.386 | 0.360 | 0.399 | 0.390 | 0.434 | 0.519 | 0.514 |
| | 192 | **0.335** | **0.370** | 0.344 | 0.375 | 0.340 | 0.368 | 0.385 | 0.401 | 0.413 | 0.418 | 0.432 | 0.453 | 0.524 | 0.526 |
| | 336 | **0.370** | **0.387** | 0.378 | 0.403 | 0.379 | 0.404 | 0.421 | 0.425 | 0.417 | 0.415 | 0.455 | 0.451 | 0.528 | 0.531 |
| | 720 | **0.417** | **0.426** | 0.426 | 0.438 | 0.426 | 0.432 | 0.494 | 0.470 | 0.460 | 0.458 | 0.562 | 0.505 | 0.551 | 0.540 |
| ETTm2 | 96 | **0.170** | **0.265** | 0.178 | 0.272 | 0.180 | 0.274 | 0.195 | 0.270 | 0.210 | 0.288 | 0.208 | 0.297 | 0.225 | 0.306 |
| | 192 | **0.222** | **0.298** | 0.230 | 0.307 | 0.233 | 0.307 | 0.257 | 0.318 | 0.273 | 0.327 | 0.284 | 0.351 | 0.290 | 0.352 |
| | 336 | **0.271** | **0.318** | 0.284 | 0.340 | 0.299 | 0.350 | 0.343 | 0.361 | 0.325 | 0.359 | 0.335 | 0.385 | 0.359 | 0.400 |
| | 720 | **0.359** | **0.365** | 0.371 | 0.392 | 0.396 | 0.420 | 0.427 | 0.419 | 0.399 | 0.407 | 0.433 | 0.431 | 0.436 | 0.433 |

### A.6 FURTHER RESULTS ON PARAMAETER SEARCH ON ETT BENCHMARK

In addition to the learning rate and sequence length parameters mentioned in the main text, we also conducted sensitivity analysis on the UniTS model with respect to parameters such as decompose kernel size, hidden size on ETT datasets and weather dataset[10]. The final results are illustated in Figure 4, Figure 5, Figure 6, and Figure 7.

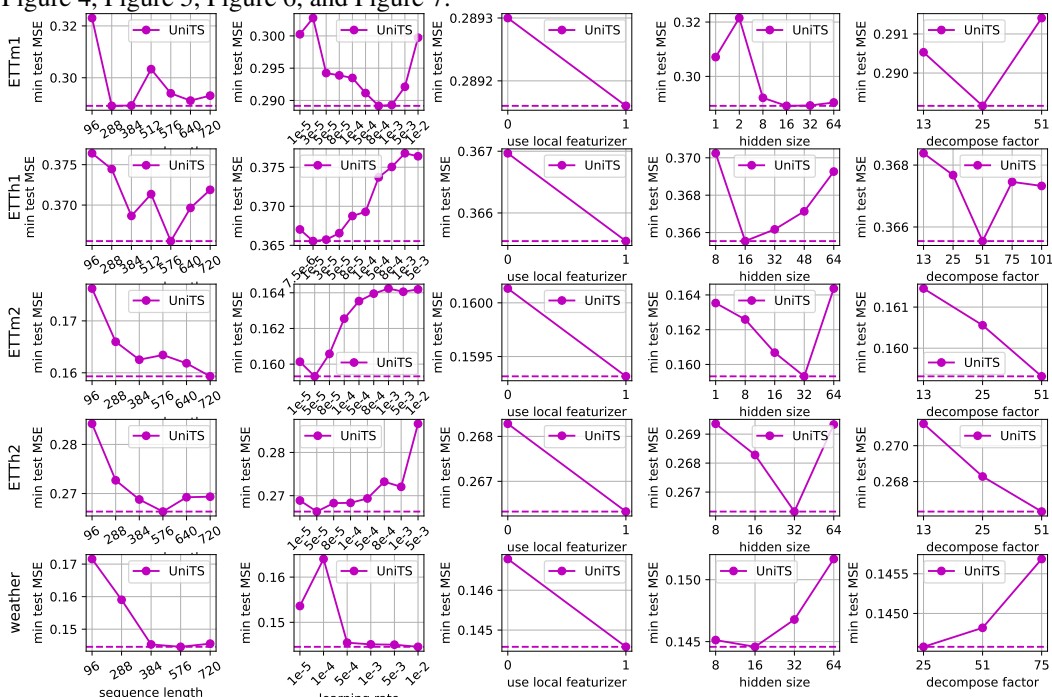

Figure 4: Performance with different parameter settings, $T = 96$

### A.7 SEARCH SPACE FOR THE HYPERPARAMTER SEARCH EXPERIMENT

The search space for the Hyperparamter Search Experiment is shown in 10.

| Parameter Name | Search Space |
|---|---|
| Lookback window size | [96, 288, 384, 576, 720] |
| Learning Rate | [1e-5, 1e-4, 5e-4, 8e-4, 1e-3, 5e-3, 8e-3, 0.01, 0.05] |
| Hidden Size | [16,32,64] |
| Use LFE | [0, 1] |
| Decompose Kernel Size | [13,25,51,75] |

Table 10: Hyperparamter Search Space.

---

[10]Due to limited computational resources, the results we have published are temporarily unable to cover the entire dataset, baseline models, and more extensive parameter searches at higher densities.

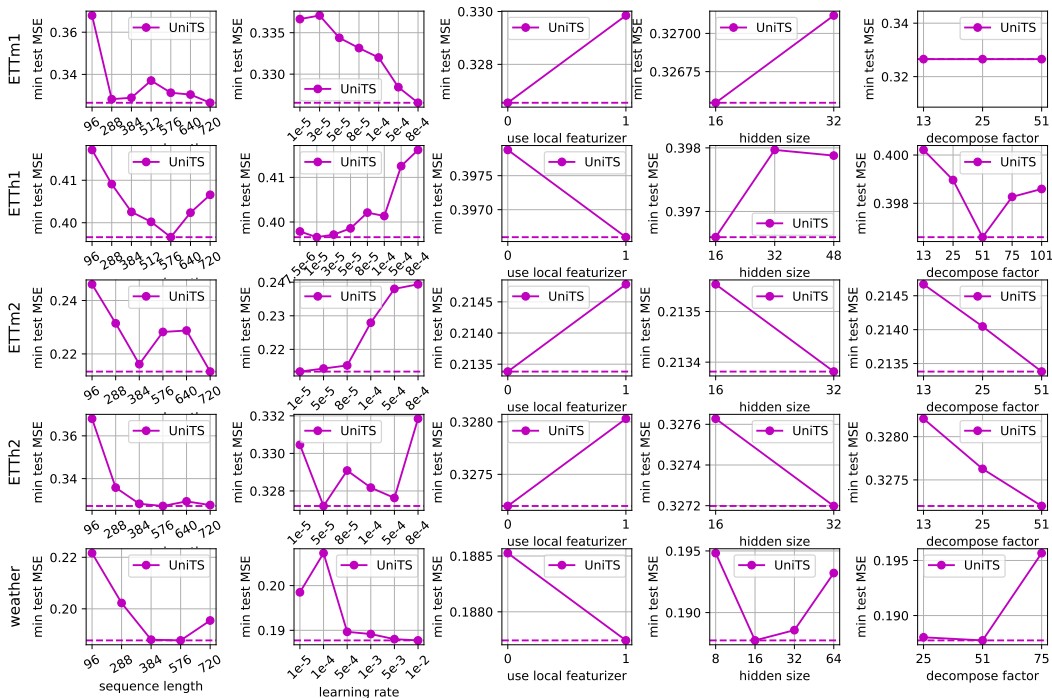

Figure 5: Performance with different parameter settings, $T = 192$

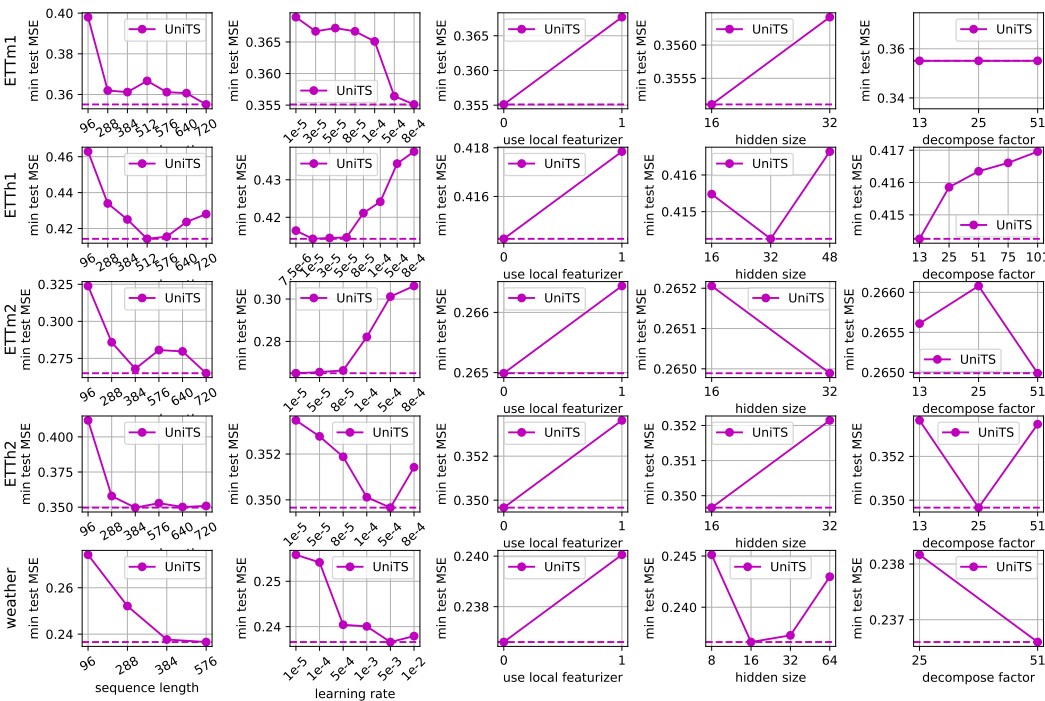

Figure 6: Performance with different parameter settings, $T = 336$.

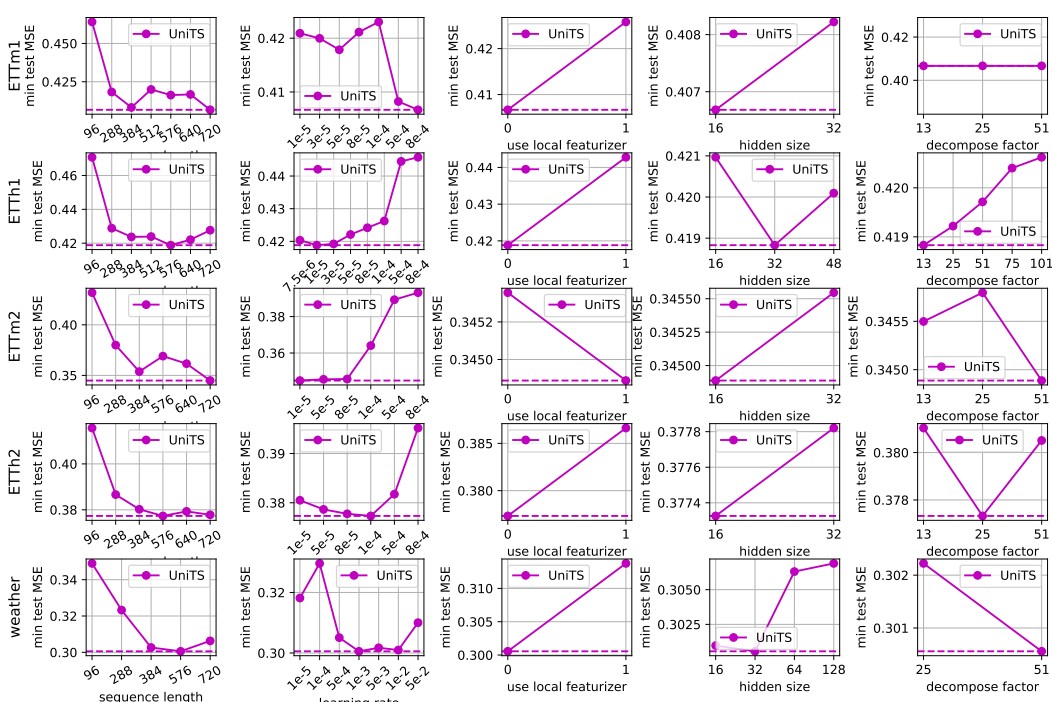

Figure 7: Performance with different parameter settings, $T = 720$.

