# OpenReview forum: "Rethinking the Temporal Modeling for Time Series Forecasting with Hybrid Modeling"
_ICLR.cc/2024/Conference — Submitted to ICLR 2024_

### Official Review · Reviewer_gsuz · 2023-10-30

**Soundness:** 2 fair
**Presentation:** 3 good
**Contribution:** 2 fair
**Rating:** 5
**Confidence:** 4

**Summary:**

This paper proposes a hybrid model that utilizes multiple structures for improved time series forecasting, demonstrating strong performance across various datasets.

**Strengths:**

1.This paper conducts comprehensive ablation experiments, analyzing the role of each module in the model. This demonstrates the importance of a multiscale hybrid model in the field of time series prediction.
2.The article provides open-source code, facilitating the reproducibility of experimental results.
3.It thoroughly compares different parameter search methods and the impact of various hyperparameter choices on the model's predictive performance.

**Weaknesses:**

1.This paper lacks innovativeness, merely combining currently high-performing models without introducing novel elements. The proposed model appears to draw heavily from existing model structures, with limited originality in its design. While the work effectively combines and leverages existing approaches, it lacks a significant level of novelty in terms of introducing truly innovative components.

2.This paper does not provide sufficiently convincing reasons for the selection of these modules.

The paper offers valuable insights into time series forecasting models and their performance. I encourage the authors to consider further enhancing the originality of the proposed model in future work.

**Questions:**

see my concerns

---

> ### Author Response · Authors · 2023-11-19
>
> We would like to express our gratitude for your patient review of our paper and the valuable problems you provided. Indeed, our model design is relatively simple, but compared to other methods that involve the coupling of multiple modules, this simplicity allows for a more straightforward evaluation of the effectiveness of each module, thereby providing a more intuitive representation of the efficacy of our model. Here is a detailed response addressing your points:
>
> 1. As we stated, this study started from the strong motivation for the decoupling of modules in previous works and a more refined analysis in previous works, as in previous studies on time series forecasting models, many modules were indeed coupled. For instance, works like Autoformer and Fedformer utilized techniques autocorrelation mechanisms, position embedding, and temporal embedding, e.t.c.; In PatchTST, mechanisms such as ReVIN, channel independence, patching, and attention mechanisms were introduced. These models are products of various coupled module methods, making it challenging to directly understand the source of performance in these models. Therefore, in our experiments, we found that comparing the performance between models with coupled modules was unfair and could lead to misleading conclusions and inaccurate model designs. We addressed this by decoupling the main modules of the models and quantitatively analyzing their impact through ablation experiments, a point not previously discussed in the literature.
>
> 2. Building on the motivation mentioned above, we identified some possibly unnecessary designs, such as the excessive pursuit of attention modeling for time series forecasting models. In comprehensive ablation experiments, we confirmed that the design like introducing context embedding and attention mechanisms might unnecessary. Furthermore, we delved deep, quantitatively and comprehensively evaluated the designs of some recent MLP models, indicating their strong performance in temporal prediction tasks. **Based on the observations mentioned above, we decoupled various modules and parameters within a hybrid modeling framework, illustrating how combining these elements and hyperparameters is a crucial theme. We appreciate your identification of shortcomings in our paper, and in the revised version, we provided a more concise and direct description to enhance readers' understanding of our contributions.**
>
> We would like to further highlight the core contributions of our paper in the revised manuscripts. And once again, thank you for your review, and we look forward to any further suggestions you may have, and we would like to express our gratitude for your thorough review of our paper and the valuable insights you provided.

---

### Official Review · Reviewer_EGjq · 2023-10-30

**Soundness:** 3 good
**Presentation:** 4 excellent
**Contribution:** 2 fair
**Rating:** 6
**Confidence:** 4

**Summary:**

Propose a long-range forecasting model for multivariate time-series using hybrid aprroach through local and global feature extraction mechanism. Their global feature mechanism leverage more like transformer like architecture and local is CNN architecture. The model adapts for input pre-precessing though instance normalization, patching, and decomposition.

**Strengths:**

1. Model Outperforms the baselines in terms of MSE, MAE score. Experiments are done in multiple public datasets including ILI, electricity, weather, traffic,etc.

2. In terms of long-range forecasting range, model outperforms with prediction length upto 60 compared to the transformer architecture and upto 720 compared to the 2nd best performing model PatchTST.

3. Authors showed an extensive ablation analysis showing usefulness of leveraging different modules (local LFE, global GFE, attention, PE, IN, etc) in the hybrid approach.

4. Paper is well-written with convincing experiments.

**Weaknesses:**

1. Is there any particular reason to not compare this model with the state-of-the-art long-range forecasting model like Spacetimeformer (Grigsby et al. 2023), and NBeats. Both models perform vey well for long range forecasting on multivariate data. SpaceTimeFormer paper shows result for prediction length upto 672 (on some weather data).

2. LFE and GFE architeture are not very novel, and mostly adapted from state-of-the-arts.

**Questions:**

1. Is there a reason why patchTST working so much better than transformer achitetures like AutoFormer/Fedformer, especially, where the ablation studies clearly show attention, PE mechanisms are useful?
2. Can you show the results compared to SpaceTimeFormer and Nbeats? SpaceTimeformer model also has local+global architeture approach.

---

> ### Author Response · Authors · 2023-11-19
>
> Thank you for your valuable feedback and constructive comments on our paper. We really appreciate the time and effort you dedicated to reviewing our work. Below is a detailed response to your points:
>
> W1: Is there any particular reason to not compare this model with the state-of-the-art long-range forecasting model like Spacetimeformer (Grigsby et al. 2023), and NBeats. Both models perform vey well for long range forecasting on multivariate data. SpaceTimeFormer paper shows result for prediction length upto 672 (on some weather data). &
> Q2: Can you show the results compared to SpaceTimeFormer and Nbeats? SpaceTimeformer model also has local+global architeture approach. & W2: Can you show the results compared to SpaceTimeFormer and Nbeats? SpaceTimeformer model also has local+global architeture approach.
>
> Ans: Thank you for your suggestion! We have supplemented the paper with additional baselines, including **SpaceTimeFormer**, **N-BEATS**, and N-BEATS' following-up work **N-HiTS**, and two new baselines which combining RevIN and linear models, as mentioned by another reviewer in **Appendix A.3**.
>
> Regarding your mention that the SpaceTimeFormer model also has a local+global architecture approach, it is worth noting that the emphasized local/global structure in the SpaceTimeFormer model is achieved by computing attention between multiple variables at each time step (local attention), while global attention is calculated directly across the entire temporal sequence. We may interpret the local attention in this approach as feature fusion performed per time step using the attention module, while global attention is similar to the attention calculation in the temporal dimension used in PatchTST. In our experimental results, we found that SpaceTimeFormer did not achieve comparable performance to UniTS and PatchTST. We attribute this to our consistent observation in experiments that the introduction of attention and context embedding did not improve model performance. Further analysis of the results in SpaceTimeFormer is needed, as they also emphasized in their paper that the introduction of the RevIN module effectively improved model performance. However, when compared with baseline models such as AutoFormer, they did not introduce the same normalization modules to these models. Taking a broader perspective, we observed that in previous works, due to the coupling of too many modules, analyzing only individual modules in experiments often cannot well explain the results, which is also a core point raised in this paper.
>
> As for the other baseline N-BEATS you metioned, we further provided the results of N-BEATS and its following-up work N-HiTS in the revised manuscript. They both adopt a design combining residual stuck with linear networks. However, these two models do not use the RevIN module, and in the final results, their predictive performance is lower than the two purely linear models we introduced, namely RMLP (RevIN+MLP) and RLinear (RevIN+Linear). Thus, this decoupling of modules in previous works and a more refined analysis are the main motivations proposed in this paper. We aim to unravel the intricacies of prior research, seeking the optimal and most suitable modules for constructing time-series prediction models. By untangling the intricacies of previous works, we hope to assist the time-series forecasting community in gaining a more fundamental and in-depth understanding of the various existing models.

---

> ### Author Response · Authors · 2023-11-19
>
> Q2: Is there a reason why patchTST working so much better than transformer achitetures like AutoFormer/Fedformer, especially, where the ablation studies clearly show attention, PE mechanisms are useful?
>
> Ans: You've raised a very good point, and indeed, as mentioned earlier, the decoupling of modules in previous works and a more refined analysis are the main motivations proposed in this paper. Both AutoFormer and Fedformer are highly innovative works, and we believe the core issue causing this confusion is: why are AutoFormer and Fedformer so effective? On this matter, we understand it is also a key argument in this paper: were these previous works truly compared fairly? For instance, when comparing AutoFormer and Fedformer with the baseline model Informer, there is a significant coupling of modules in the comparison between models. This fails to reflect which modules truly influence the model and contribute to performance improvements/degradations. In our experiments, we observed that the inclusion of positional encoding actually leads to a decline in model performance. Furthermore, we may mention that the recent and representative work DLinear, that did not use positional encoding modules, achieved significantly better performance than AutoFormer and Fedformer on time series forecasting tasks using only a simple linear model. Therefore, we posit that the perceived effectiveness of position encoding in AutoFormer is, to a large extent, due to insufficient testing of its parameters and modules (we believe the effectiveness of positional encoding is not discussed by the authors in the original paper). The coupled modules contribute to the perception of their effectiveness, which, in reality, may not be the case.

---

### Official Review · Reviewer_Y4Q5 · 2023-10-31

**Soundness:** 2 fair
**Presentation:** 3 good
**Contribution:** 3 good
**Rating:** 6
**Confidence:** 5

**Summary:**

This paper proposes to combine different models for the long-term time series forecasting in a framework "UniTS", in which Convolutional Neural Network is used to extract local feature and MLP (or Transformer) is used to extract global feature. In addition, this paper points out the unfair comparison issue in existing works due to the lack of standardized parameter design. Experiments on the 8 benchmark datasets are conducted to evaluate the proposal.

**Strengths:**

This paper uses the advantages of different models and combines them in a framework for a better long-term time series forecasting. This paper also points out the unfair comparison issue in existing works due to the lack of standardized parameter design.

**Weaknesses:**

1. Some parts of the proposal UniTS are not introduced clearly. e.g.,

What are the meanings of H^{l,N} and H^{g,N} in Figure 1?

How to do the LFE and GFE after Patching?

What is the meaning of j and what is the Decompose in the first equation of section 3.2.3?

2.  The RLinear and RMLP models in the following reference outperform PatchTST on some datasets. It is better to compare with them as well.

Li, Zhe, et al. "Revisiting Long-term Time Series Forecasting: An Investigation on Linear Mapping." arXiv preprint arXiv:2305.10721 (2023).

3. I think it is unfair to compare with PatchTST/64 which uses lookback window length 512 only. As shown in the Figure 2 of the PatchTST paper, the performance is also changed with different lookback windows. It is better to choose the best results from different lookback windows for PatchTST as well, since this paper highlights the unfair comparison issue.

4. The hyperparameter selection (including learning rate, hidden size, e.t.c. besides the use of lookback window length) is only used for the proposed UniTS but not for other baselines. I am doubting whether it results in another unfair comparison problem.

5. There is no complexity analysis, especially, the complexity introduced by the hyperparameter selection.

Typos:
"four primary categories" in Page 1 -> "three primary categories"

"Table 1 provides a overaTidell view analysis" in Page 6 -> "Table 1 provides a overall view analysis"

V_i is not used before explaining it in Page 8.

**Questions:**

Same to the Weaknesses.

---

> ### Author Response · Authors · 2023-11-19
> **1/2**
>
> Dear Reviewer,
>
> Thank you for your thorough review and valuable feedback on our paper. We appreciate your insights and have carefully considered your suggestions. Here is our response to address the key issues you raised:
>
> Q1: Some parts of the proposal UniTS are not introduced clearly
>
> Ans:We will provide additional clarity on the meanings of H^{l,N} and H^{g,N} at the caption of Figure 1, ensuring a more comprehensive explanation in the revised manuscript. Regarding how the data is passed to the GFE and LFE modules after patching, we have provided a connecting description in the main text. You can find it highlighted in blue in section 3 of the revised manuscript. Additionally, we have further corrected the typos and wording issues in the original text that you mentioned, which could potentially be misleading.
>
> Q2: The RLinear and RMLP models in the following reference outperform PatchTST on some datasets. It is better to compare with them as well.
>
> Ans: In light of your and another reviewer's suggestions, we have incorporated the results of RLinear, RMLP, and two additional baselines in the revised version of the paper (**Appendix A.3**). Both RLinear and RMLP underscore the effectiveness of combining ReVIN with linear networks, showcasing superior performance to PatchTST on widely used datasets. These findings align seamlessly with our experimental observations. The additional experimental results have been appended, and we present them in tabular format for your convenience.
>
> Q3:I think it is unfair to compare with PatchTST/64 which uses lookback window length 512 only. As shown in the Figure 2 of the PatchTST paper, the performance is also changed with different lookback windows. It is better to choose the best results from different lookback windows for PatchTST as well, since this paper highlights the unfair comparison issue.
>
> Ans: Thank you very much for pointing out the issues covered in the description of our paper. Indeed, in our experiments, we performed parameter search for each baseline model as well. For DLinear and PatchTST, we essentially reproduced the results reported in their respective papers. However, as you rightly pointed out, directly using values such as the lookback window size of 512 from the original PatchTST paper could lead to suboptimal results and raise concerns about fairness. Thus, it is essential to clarify this aspect, as it directly relates to the fairness of performance comparisons between various models.In the revised manuscript, we use the optimal results obtained through parameter search in our own replication as the results in Table 1. We have also clarified how the data in Table 1 was obtained in section 4.1 to ensure a more fair perspective for the result studies of our experiments.

---

> > ### Comment · Reviewer_Y4Q5 · 2023-11-21
> >
> > Thank authors for the response. For the LFE and GFE after Patching, I think the circles in Figure 1 mean time steps instead of patches, so I am still not sure how to process patches in LFE and GFE.

---

> > > ### Author Response · Authors · 2023-11-22
> > >
> > > Dear Reviewer,
> > >
> > > Hello, and we sincerely apologize for any confusion our paper may have caused you. Your observation is correct; the dots in the figures represent time steps. We acknowledge the lack of detailed explanation regarding the UniTS model in the previous version, and we appreciate your understanding. After the patching operation, we pass the patches into GFE and LFE.
> > >
> > > In GFE, each patch undergoes preliminary feature extraction through one (or more) linear layers. Subsequently, the features of all patches are concatenated and passed through another linear layer to output the global feature embedding. In LFE, features of each patch are extracted using (a) 1D convolutional kernel(s), and the results are concatenated to form the local feature embedding.
> > >
> > > Besides, we provide further clarification here to explain why we choose to perform feature extraction after patching rather than directly extracting features from the entire sequence.
> > >
> > > 1. One of our main focuses is on comparing with the representative work "PatchTST" related to transformers. For datasets with a large number of data, such as traffic (862 series) or ECL (321 series), directly calculating the attention over the entire time series without patching incurs high computational complexity. To quantitatively test the effectiveness of the attention module, we adhere to this practice in the overall design;
> > > 2. In our experimental observations and observations in works such as PatchTST, using patched sequences does not degrade forecasting performance. Moreover, as it focuses on computing each patch rather than the entire sequence, it allows for lower model complexity. For instance, in our experiments with reduced hidden size (e.g., selecting hidden size $d$ as 16), both the baseline model and UniTS exhibit stable performance. The parameter count remains low, and in some datasets, we can compress it to 10k parameters. As a comparison, using a direct linear mapping like DLinear with an input of 336 and an output of 96 already exceeds 30k parameters, while our global feature extractor module with a hidden size of 16 requires only around 10k parameters, achieving excellent performance with fewer parameters than a direct linear mapping.
> > > 3. Then, for uniformity in data dimensions, we also use patched sequences for feature extraction in LFE;
> > >
> > > Again, thank you for your valuable feedback, and we have incorporated these details in section 3.2.2 and section 3.2.3, and we have updated Figure 1, indicating that in both GFE and LFE, the processing involves patched sequences, and their corresponding operations at output.
> > >
> > > Best,

---

> > > > ### Comment · Reviewer_Y4Q5 · 2023-12-05
> > > >
> > > > Thank authors for the further response. I updated my rating. However, I still think it is better to consider the fair comparison with baselines more deeply.

---

> ### Author Response · Authors · 2023-11-19
> **2/2**
>
> Q4&Q5:The hyperparameter selection (including learning rate, hidden size, e.t.c. besides the use of lookback window length) is only used for the proposed UniTS but not for other baselines. I am doubting whether it results in another unfair comparison problem; There is no complexity analysis, especially, the complexity introduced by the hyperparameter selection.
>
> Ans:Your observation on the necessity of discussing complexity is well-founded. In our paper, In our discussions, we proposed a method that essentially decomposes modules previously coupled in various temporal forecasting models. We discuss which of these modules are useful and how the choice of modules affects the model. This makes parameter selection, previously overlooked in the literature, an unavoidable issue. In previous discussions, the performance of models presented is often the result of manual selection on the given architecture, making it a core issue in our paper.
>
> Analyzing this issue directly is challenging for two reasons: first, we cannot exhaustively explore all possibilities for each model, and second, conducting a parameter search in a given parameter space would introduce human bias, leading to potential unfairness. However, based on their performance on real-world datasets, we can propose a comparison method in limited scenarios. In practice, the most effective time series prediction models do not employ overly complex network structures or high-dimensional parameters. Through our experiments, we observed that overly complex parameterized structures often have a negative impact on performance.
> From the perspective of the dataset used and the models employed in this paper, if our goal is to find an optimal model, we can define a search space. In actual experiments, we found that for almost all models, we only need to search within a relatively small hidden size and a shallow network range. Therefore, for all models, the number of times they select optimal network parameters is relatively small. From this perspective, in cases where the search space difference is not significant, in terms of computational complexity, the cost of parameter search mainly comes from the training cost of each parameter configuration, which can be further quantified by the training cost per epoch for each model. From this point, we provide a further discussion of model complexity in **Appendix A.4** in the revised manuscript.
>
> It is important to note that the comparison we provide is not perfect and does not represent absolute fairness. However, we have demonstrated that this search is simple and feasible within the existing knowledge. In a limited time cost, using the structure we propose, we can train models that achieve effective time series prediction performance.

---

### Author Response · Authors · 2023-11-19
**Appreciation for valuable Feedback from Reviewers**

Dear Reviewers,

Thank you for your thorough review and valuable feedback on our manuscript. In response to your suggestions, we have incorporated additional baselines into our study. **Over the next few days, we will be uploading organized code to facilitate the reproduction and validation of results with the new baselines. Given the time constraints for the revision, we will be consistently updating the content in the revised manuscript.**

If you have any further comments or concerns regarding our modifications, please feel free to communicate with us. We appreciate and value your feedback, ensuring that our paper reaches the better quality. Thank you once again for your patience and support in reviewing our work.

Best,

---

### Author Response · Authors · 2023-11-23

Dear Reviewer,

We greatly appreciate your initial valuable comments. We hope that you could have a quick look at our responses to your concerns. It would be highly appreciated if you could kindly update the initial rating if your questions have been addressed. We are also happy to answer any additional questions before the rebuttal ends.

Best Regards,

---

### Meta-Review · Area_Chair_v9Ly · 2023-12-05

**Metareview:**

This paper proposes a hybrid model for long-range forecasting of multivariate time series data. The model utilizes a combination of local feature extraction (LFE) and global feature extraction (GFE) mechanisms, with the LFE employing a convolutional neural network (CNN) architecture and the GFE leveraging a transformer-like architecture.

The paper presents extensive experiments on multiple public datasets, demonstrating its superior performance over baseline models in terms of MSE and MAE scores. It further highlights the model's superior long-range forecasting capability. The paper is well-written and provides convincing evidence of the model's effectiveness.

However, the paper also raises some concerns. Firstly, it does not compare the proposed model with all notable state-of-the-art long-range forecasting models. Secondly, the paper lacks significant originality, as the LFE and GFE architectures are largely adapted from existing models. While the model effectively combines these approaches, the reviewers cannot support that it has truly innovative components. Additionally, the rationale for selecting these specific modules requires further explanation.

Overall, the paper presents a valuable contribution to the field of time series forecasting. It demonstrates the potential of a multiscale hybrid approach and provides insights into the effectiveness of different modules. However, further work is needed to address concerns regarding model novelty and justification for design choices.

**Justification For Why Not Higher Score:**

- Innovativeness is a concern, similar modules have been proposed
- More comprehensive analyses and clear motivations on the proposed modules
- More complete benchmarking with the recent state-of-the-art
- Paper writing and math notations can be improved

**Justification For Why Not Lower Score:**

- Promising results across multiple benchmarks

---

### Decision · Program_Chairs · 2024-01-16

Reject